# Effective plans for hospital system response to earthquake emergencies

Luis Ceferino [1,2 ✉], Judith Mitrani-Reiser[3], Anne Kiremidjian[1], Gregory Deierlein[1] & Celso Bambarén[4]

Hospital systems play a critical role in treating injuries during disaster emergency responses. Simultaneously, natural disasters hinder their ability to operate at full capacity. Thus, cities must develop strategies that enable hospitals' effective disaster operations. Here, we present a methodology to evaluate emergency response based on a model that assesses the loss of hospital functions and quantifies multiseverity injuries as a result of earthquake damage. The proposed methodology can design effective plans for patient transfers and allocation of ambulances and mobile operating rooms. This methodology is applied to Lima, Peru, subjected to a disaster scenario following a magnitude 8.0 earthquake. Our results show that the spatial distribution of healthcare demands mismatches the post-earthquake capacities of hospitals, leaving large zones on the periphery significantly underserved. This study demonstrates how plans that leverage hospital-system coordination can address this demand-capacity mismatch, reducing waiting times of critically injured patients by factors larger than two.

[1] Department of Civil and Environmental Engineering, Stanford University, Stanford, CA, USA. [2] The Andlinger Center for Energy and the Environment, Princeton University, Princeton, NJ, USA. [3] Materials and Structural Systems Division, Engineering Laboratory, National Institute of Standards and Technology, Gaithersburg, MD, USA. [4] School of Public Health and Administration, Universidad Peruana Cayetano Heredia, Lima, Peru. ✉email: lceferinor@gmail.com

Hospital systems are at the core of disaster resilience because they must provide timely critical healthcare services to communities during an emergency response. Because cities are becoming larger and more densely populated, natural disasters are impacting public health on a larger scale. A database including the most 21,000 devastating disasters worldwide since 1900 indicates that 50% of disasters with the largest number of injuries occurred only during the last 20 years[1]. Natural disasters such as earthquakes, landslides, floods, typhoons put heavy demands on hospital systems because these disasters can cause thousands or even tens of thousands of injuries in a short timespan. At the same time, natural disasters cause massive disruptions to hospital systems by damaging their supporting infrastructure. For example, the M 7.6 1999 Turkey earthquake caused ~50,000 injuries in Izmit and disrupted 10 major hospitals, which required the relocation of most patients from these hospitals[2].

Natural disasters also demand large mobilizations of patients. For example, 2k to 3k patients had to be transferred between hospitals after the M 8.8 2010 Chile earthquake[3]. In large urban centers, such massive mobilization of patients requires an organized and system-level response to treat them timely. Yet, evidence shows that responses are often not the result of a global, system-level strategy but rather local and haphazard[4]. After the M 6.7 1994 Northridge earthquake, two hospitals transferred their patients to a non-functional hospital, thus the patients had to be transferred a second time[5].

Recognizing the importance of organized and system-level hospital response, the World Health Organization (WHO) and Pan-American Health Organization (PAHO) urge countries to institute policies to strengthen capacities and enhance coordination in the hospital system to make efficient use of resources at national and regional levels during emergency response[6,7]. To effectively develop measures for capacity-enhancing prioritization and resource sharing and allocation, national and regional governments require information based on robust methodologies that can characterize hospitals' emergency response as an interconnected system rather than as isolated units. However, research has not yet focused on developing such system-level methodologies that can support governments to elaborate effective plans for emergency response and quantify their potential benefits in treating patients more effectively and saving lives.

Instead, recent studies have primarily focused on modeling emergency response only on a single-hospital scale. Some studies rely on disaster analytics to evaluate post-disaster functionality of the supporting infrastructure in the individual hospitals[8–11]. Other studies use emergency medicine modeling tools, such as discrete event simulation and flow models, to characterize emergency response and evaluate post-disaster resource allocation but also on a single-hospital scale[12–15]. Lack of methods and high-resolution disaster risk data have hindered the extension of single-hospital scale analyses to system-level analyses on an urban scale. As a result, regional emergency response plans have not effectively addressed capacity-enhancing prioritization and resource sharing and allocation in hospital systems, especially in large and complex urban centers.

To demonstrate the benefits of organized, system-level responses, we present findings from applying a methodology that characterizes the disaster emergency response of hospitals as a system. Our integrative methodology combines models of multiseverity earthquake casualty estimation[16,17] and post-earthquake hospital functionality with a proposed network flow model. The network flow model captures the system behavior of emergency responses through the evaluation of patient treatment, triage processes, and patient transfers across a large number of hospitals, which establishes a fundamental difference from existing formulations focused on analyses on a single-hospital scale.

We apply the methodology to the city of Lima, Peru, subjected to a M 8.0 earthquake scenario. We selected Lima because it has a high seismic risk and it has recently built a unique data set containing high-resolution hospital vulnerability. We use city-wide data on the seismic vulnerability of >1.5 M buildings in Lima to estimate casualties and data including the seismic vulnerability of 41 public hospital campuses (composed of +700 buildings) and their respective operating rooms and ambulance resources[11,18,19].

We propose a metric based on patient waiting times and effective use of ambulances as a performance measure for developing emergency response plans. Our focus is on high-severity injuries that require surgical procedures in operating rooms. We evaluate the spatial distribution of high-severity injuries in the city at a high spatial resolution of 1 km. Then, we compare the spatial distribution of casualties with the distribution of functional operating rooms in the hospital system, identifying the zones more likely to be underserved during the emergency response.

Combining the network flow model with an optimization formulation, we assess the performance of four alternative emergency response plans to treat the patients in the city. The first and second emergency plans are baseline strategies with limited levels of coordination that enable hospitals to respond mostly locally. The third and fourth emergency plans are strategies with higher levels of coordination at the system level and that use our proposed formulation to optimize the effectiveness of the response. In both strategies, hospitals respond as a system by sharing their ambulances among themselves to transfer patients according to post-earthquake needs. With the fourth strategy, emergency medical teams (EMTs) supply the system with additional mobile operating rooms in key locations in the city. Through the Action Plan for Humanitarian Assistance, the WHO and PAHO require countries to elaborate policies for deploying EMTs to assist people affected by emergencies and disasters[20], thus, this study aims to directly inform policies for EMT deployment in countries with high seismic risk.

Our results show the improved performance of the third and fourth strategies owing to their high-coordination capacities leading to shorter patient waiting times and more effective ambulance usage and patient transfers. We also identify the most important roads for patient transfers, the ones that connect zones with lower hospital capacity to zones with higher capacity in the city. This research represents a first-cut assessment of the effectiveness of emergency response policies to inform city-scale decision-making aiming to more effective treatment of patients during an emergency response to a major earthquake.

## Results

**Earthquake scenario.** Lima is a large city with a population close to 10 million people, where previous large earthquakes[21] have caused large numbers of casualties[22]. In our case study, we assess an earthquake scenario simulated according to the seismotectonics of the 1940 M 8.0 earthquake, which occurred in close proximity to Lima[23]. Figure 1 shows the estimated rupture area of the 1940 earthquake and its proximity to the city. Our methodology estimates the impact of this disaster scenario on the demands on healthcare by quantifying earthquake casualties and on the capacity of healthcare by quantifying the post-earthquake reduction in functionality in the hospital system.

**Earthquake casualties.** We estimate that on average close to 4.7k people will require surgical procedures in operating rooms after

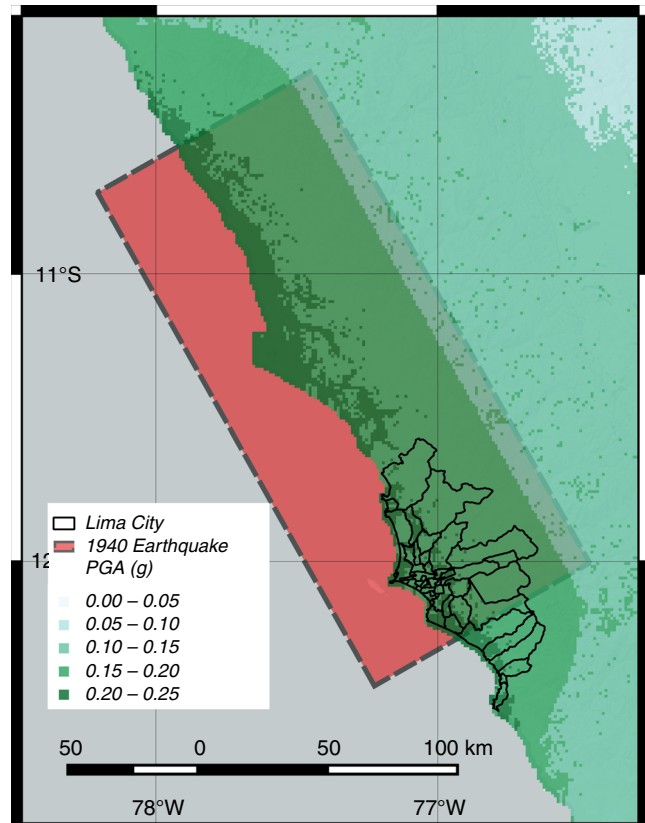

**Fig. 1 Earthquake scenario representing the M 8.0 1940 earthquake in Lima.** The earthquake occurred in the subduction fault in the coast of Lima and caused widespread damage to the city[22,70]. The estimated area of fault rupture is shown in red. The edge dimensions were estimated with empirical formulas[63]. The fault plane dips 15°, where the edge underneath the coast is deeper than the edge under the ocean. The median peak ground acceleration (PGA) is also estimated with empirical formulas[64]. The shaking attenuates for regions further away from the rupture in the fault plane. Lima city and its districts are delimited by the black shapes. Source data are provided as a Source Data file, and the base map layer is available under a https://www.openstreetmap.org/copyright Open Database Licence (© OpenStreetMap Contributors).

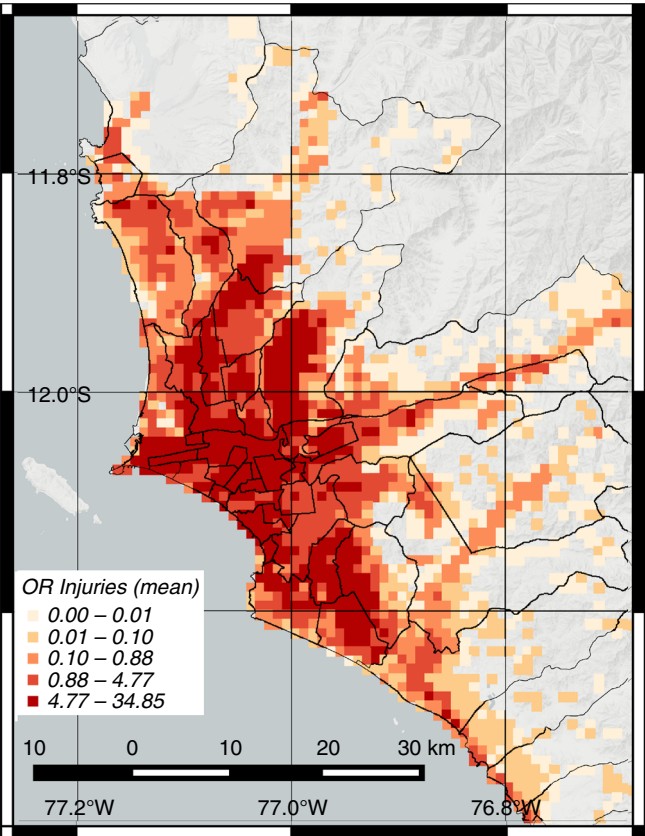

**Fig. 2 Casualty scenario for M 8.0 earthquake occurring at nighttime in Lima.** The plot shows the spatial distribution in km² of the mean number of earthquake injuries requiring surgical procedures after the M 8.0 seismic event. The intervals represent quantiles (20th-percentile increments) on the spatial data. Source data are provided as a Source Data file, and the base map layer is available under a https://www.openstreetmap.org/copyright Open Database Licence (© OpenStreetMap Contributors).

the M 8.0 earthquake. This estimate results from applying a probabilistic model that utilizes high-resolution building seismic vulnerability data, population distribution, and soil conditions to evaluate multiseverity earthquake casualties caused by widespread building damage[16,17] (see "Methods"). Earthquake injuries can have different severity degrees, ranging from small bruises to more serious spinal cord injuries[24,25]. The 4.7k patients requiring surgical procedures will have high-severity injuries such as compound bone fractures, punctured organs, or crush syndrome with open wounds, thus needing timely interventions for stabilization and treatment.

Our results are designed for an earthquake occurring at nighttime. This scenario represents a potential worst-case scenario because it is when most people are inside residential buildings, which are particularly vulnerable in Lima. Predominantly, the city's periphery has vulnerable residential infrastructure as a result of poor construction practices and lack of seismic code enforcement[26,27]. Fig. 2 shows the spatial distribution of the average number of patients who will require surgical procedures for the nighttime scenario. A comparison with the spatial distribution of nighttime population density in Lima (Supplementary Figure 1) indicates that many of these patients are

located in high-density zones. Because earthquakes have a uniform distribution of occurrence during a day, this scenario has a slightly higher likelihood of occurring than the other two important earthquake scenarios to consider: commuting hours from 6 am to 8 am and from 4 pm to 8 pm, and working hours from 8 am to 4 pm[16,17]. The Methods section includes a description of the data requirements to fully evaluate scenarios at other times with our formulation, and the Supplementary Information includes a discussion of their potential results (see Supplementary Figures 2 and 3).

**Post-earthquake hospital capacity.** We estimate that on average only 87 of 182 total hospital operating rooms (48%) will be functional after the M 8.0 earthquake. We verified that our results were consistent with the reductions of hospital capacities in past earthquakes. For example, similar to our predicted high levels of capacity reduction, the M 8.0 2007 Pisco earthquake in Peru reduced the hospital bed capacity to 38%[28]. Our estimates of capacity reduction result from performing a probabilistic earthquake simulation on a high-resolution data set (see Methods). The data set includes the structural vulnerabilities of +700 buildings belonging to 41 healthcare campuses[18], the operating room resources, and Hospital Safety Index (HSI) of each campus. HSI is a metric created by WHO to measure post-disaster functionality potential due to multiple factors such as backup water,

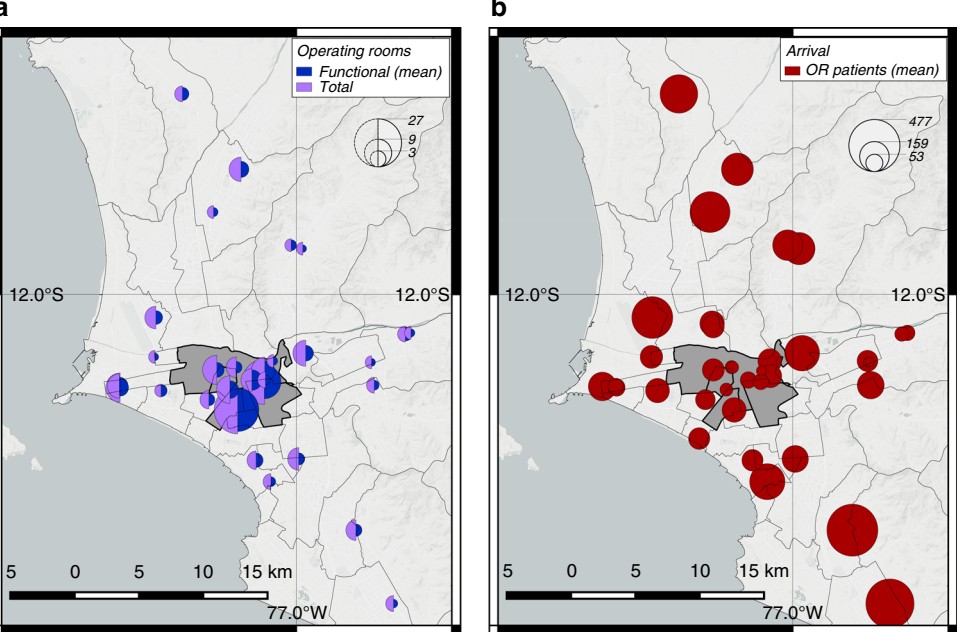

**Fig. 3 Distribution of operating rooms and patient arrivals in Lima, where the circle sizes represent their relative values. a** Current number of operating rooms in hospital locations[18] and mean estimates of functional operating rooms after the M 8.0 earthquake. **b** Mean estimates of total arrivals of patients who will need surgical procedures after the earthquake. The comparison of both plots shows that the capacity of the system is heavily centralized, whereas the demand is concentrated in the periphery of the city. Source data are provided as a Source Data file, and the base map layer is available under a https://www.openstreetmap.org/copyright Open Database Licence (© OpenStreetMap Contributors).

power, medical resources, and hospital accessibility[29]. This unique data set in combination with the earthquake simulation enables us to capture the residual hospital functionality on a large urban scale.

Fig. 3a shows the spatial distribution of both the operating rooms in the data set and the average predictions of operating rooms after the earthquake. Both spatial distributions are heavily uneven across the city. In the data set, 95 operating rooms (52%) are concentrated in only four centric districts, Lima, Breña, La Victoria, and Jesús María, whose summed areas represent <2% than the total area of the city. The earthquake slightly worsens such a resource centralization owing to the non-uniform spatial distribution of earthquake shaking and the variations in the vulnerabilities of hospitals' buildings according to their construction age and standards or structural types (see Supplementary Figure 4). As a result, we estimate that these four districts will have on average 49 functional operating rooms, 57% of the total functional operating rooms, during the emergency response.

**Demand-capacity mismatch of health services**. We analyzed patient arrivals to the hospitals and found that patient distribution significantly mismatches the distribution of residual hospital resources after the earthquake. We assumed that search and rescue (SAR) teams, relatives, friends, and neighbors will initially transport patients to the triage areas in the closest hospitals as it occurred after previous earthquakes[30]. Fig. 3b shows the distribution of the total arrivals of patients who will need surgical procedures in each hospital. In contrast to the distribution of functional operating rooms, injuries are mainly located in the periphery. Only 596 patients would arrive at the hospitals in the four centric districts highlighted in Fig. 3b. Those patients represent only 13% of the total demand for surgical procedures. However, as described earlier, these four districts will concentrate 57% of the functional operating rooms available.

Such a mismatch in the distribution of demands and capacities creates localized health service imbalances, leading to long patient

waiting times, with particularly severe effects in the periphery. The mismatch can become even more critical because the number of injuries needing surgical procedures and the functional operating rooms are negatively correlated. Our findings show a strong correlation (−0.45) between the simulations of earthquake injuries and functional operating rooms across the system in the city (Fig. 4). Such a large correlation indicates that an earthquake that injures a larger amount of people will likely be very destructive; thus, that scenario will also cause a heavier disruption to the hospital system.

Emergency plans can play a key role in addressing this mismatch and improve the treatment effectiveness in the city if they either mobilize patients from lower-capacity zones to higher capacity zones or supply lower-capacity zones with additional resources. We tested four emergency plans. Two of them are baseline strategies that only require limited coordination in the system, whereas the other two are strategies that require higher coordination at the system level. To evaluate their performance during the emergency response, we used a system metric based on city-wide patient waiting times and effective use of ambulance resources (see "Methods"). Waiting times[31] are a key metric to establish necessary patient stabilization procedures until there is an available operating room in the queue, and the use of ambulances is a complementary metric to ensure that patient transfers occur effectively.

**Baseline Strategies with Limited Coordination**. In the first strategy, hospitals send patients to the closest hospital with functional operating rooms only if all their operating rooms are non-functional after the earthquake. In this strategy, hospitals use their own ambulance resources to transfer their patients. This strategy only requires limited coordination between pairs of hospitals located relatively close to each other, representing an emergency response where the system becomes a set of islands composed of districts or neighborhoods that treat injuries independently of each other. With this strategy, our mean estimates

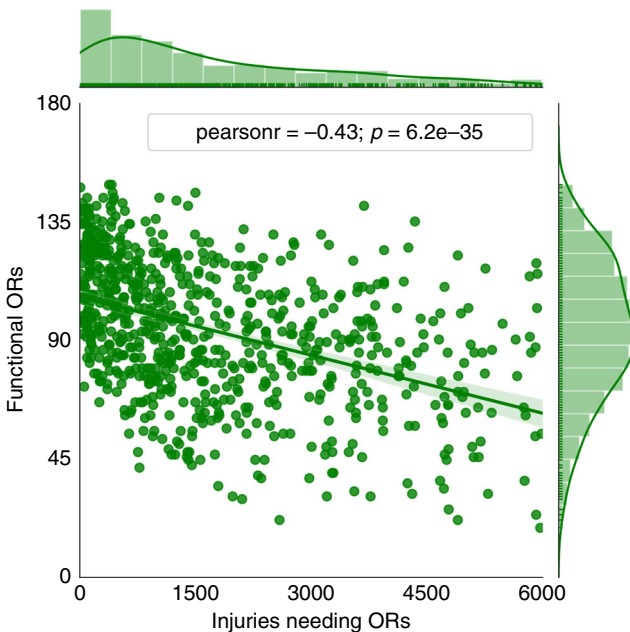

**Fig. 4 1000 simulations of number of casualties needing surgical procedures in operating rooms (ORs) and number of functional operating rooms after the M 8.0 earthquake.** The simulations result from probabilistic earthquake modeling (see "Methods") and capture uncertainty in ground shaking, building damage, injury occurrence, and hospital functionality. The linear trend indicates a negative correlation between the functional ORs and the number of injuries in the simulations. Source data are provided as a Source Data file.

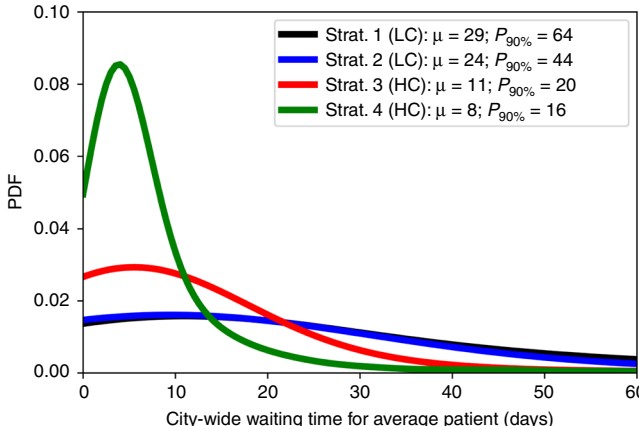

**Fig. 5 Distribution of city-wide average waiting time for treatment after the earthquake according to different emergency response plans, highlighting mean ($\mu$) and 90th-percentile values ($P_{90\%}$).** The time is measured from when the patient is injured by the earthquake until he or she is treated in an operating room. Strategies 1 and 2 are baselines with limited coordination (LC) capacities, whereas strategies 3 and 4 introduce higher coordination (HC) capacities across the whole system level for resource allocation and patient transfers. The ambulance usage and the treatment spatial distribution are show in Fig. 6 for each plan. Source data are provided as a Source Data file, and the base map layer is available under a https://www.openstreetmap.org/copyright Open Database Licence (© OpenStreetMap Contributors).

indicate that the average waiting time will be 29 days to receive treatment in operating rooms (Fig. 5). In some worst-case scenarios, this metric could even increase up to 64 days, as indicated by the 90th-percentile of this distribution. The 1988 Armenia earthquake showed that delayed surgical treatment can worsen the patients' health to life-threatening conditions[32], for example, those who have intra-abdominal hemorrhage or need emergency amputation. Thus, such long waiting times in Lima can result in many additional deaths.

Fig. 6a shows the mean estimates of the spatial distribution of treated patients at each hospital and the patient transfers between hospital pairs. Though this strategy can offload demands in critical zones, it does not effectively mobilize patients from lower-capacity zones to higher capacity zones. Hospitals with more functional operating rooms treat a similar number of patients as hospitals with fewer operating rooms. Hospitals in the four centric districts highlighted previously only treat 1.3k patients, which represents 27% of the demand for operating rooms, despite having 57% of the total capacity. In addition, because hospitals do not share ambulance resources, we find that ambulances are the bottleneck of the system in the periphery. If hospitals do not have sufficient ambulance resources, their patients will lose the opportunity to be treated more promptly in other less crowded hospitals.

In the second strategy, hospitals send patients to the hospital with the largest number of functional operating rooms. With this strategy, hospitals send patients only if all their operating rooms are non-functional and use their own ambulance resources. The mean and the 90th-percentile estimates of city-wide waiting time are 24 and 44 days, respectively, outperforming the first baseline strategy, but not significantly (Fig. 5). With this strategy, the system mostly relies on the largest two hospitals, located in the highlighted centric districts, to meet the demands of surgical

procedures. Fig. 6b shows the corresponding distributions of treated patients and transfers. The two largest hospitals treat 2.6k patients, 54% of the total demand, though their functional operating rooms only constitute 34% of the total. Because multiple hospitals with non-functional operating rooms send patients to the same large hospitals under this strategy, their operating rooms overflow. Moreover, such a strategy leads to heavy use of roads from the periphery to the city center. For example, our mean estimates indicate that 177 patients would have to be transported from the southernmost hospital alone to the largest hospital, nearly twice as many as the maximum number of transfers between any hospital pair in the first baseline strategy.

**Sharing ambulances.** In the third strategy, hospitals effectively transfer patients across the system (see "Methods"). They are able to share their ambulance resources across the system. This strategy represents an emergency plan that requires high coordination at the system level. Using this strategy, the mean and 90th-percentile estimates of city-wide waiting time are 11 and 20 days, respectively, significantly outperforming the first and second strategies by factors of 2.6 and 3.2 in the mean estimates, respectively (Fig. 5). Because an optimization formulation is used at the system level under this policy, patients are effectively transported patients from lower-capacity zones to higher capacity zones, leading to a more effective use of the functional operating rooms across the city.

The distribution of treated patients matches the distribution of the residual operating room capacity in the system (Fig. 6c). Around 2.5k people are treated in the hospitals in the four highlighted centric districts, which represents 53% of the total patients in the city and closely approaches the residual capacities in this zone (57%). Unlike the second strategy, this strategy does not overload the capacities in the two largest hospitals by sending most patients to them, instead it distributes patients across the system according to the residual capacities of each hospital. In

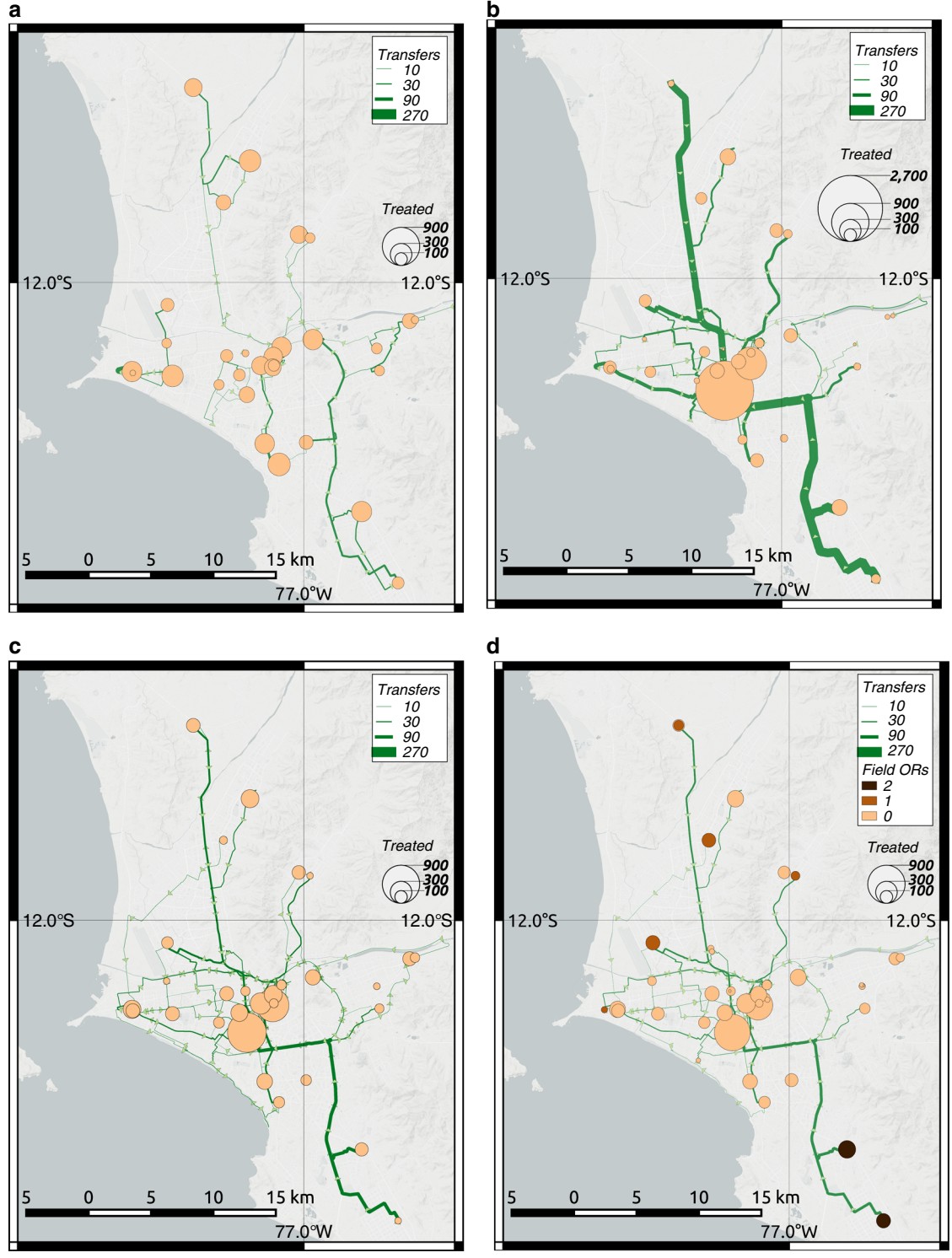

**Fig. 6 Spatial distribution of average treated patients and patient transfers in the hospital system for four different emergency response plans. a** Strategy 1. **b** Strategy 2. **c** Strategy 3. **d** Strategy 4. The figure only shows roads between hospital pairs that transferred at least five patients. Source data are provided as a Source Data file, and the base map layer is available under a https://www.openstreetmap.org/copyright Open Database Licence (© OpenStreetMap Contributors).

addition, unlike with the first strategy, ambulance capacities do not bottleneck the system with this policy. Because hospitals share all ambulances in the system, the ambulances will work where they are most needed, traveling between the periphery and the city center. An emergency plan that implements such a strategy will lead to a more balanced use of ambulances throughout the city and thus it will offload critical roads. With this strategy, close

to 76 patients would have to be transported from the southern-most hospital to the largest hospital, less than half the number with the second baseline strategy.

**Deployment of additional operating rooms by EMTs.** In the fourth strategy for effective emergency response, EMTs will

deploy 15 additional mobile operating rooms to alleviate high demand-capacity gaps across the system (see "Methods"). We assume that the additional operating rooms will be functioning 3 days after the earthquake. Though 3 days can be a short time to deploy additional operating rooms, well-organized surgical teams have demonstrated that they can deploy their medical resources even within 48 hours after the disaster even from abroad, e.g., 2007 Pisco earthquake in Peru[33]. This strategy deploys the operating rooms close to existing hospitals to leverage their triage areas and additional resources such as personnel, power generators, or backup water. Such arrangements worked well during the emergency response following the 2010 Chile Earthquake and extended the existing capacity of healthcare facilities[3]. In addition, with the fourth strategy hospitals can share ambulance capacities across the city, analogously to the emergency response with the third strategy. Because an optimization formulation is also used, the additional operating rooms are effectively deployed in locations and quantities that are critical to improve the performance of the emergency response. Using this policy, the mean and 90th-percentile estimates of city-wide waiting times are 8 and 16 days, respectively (Fig. 5). As expected, these estimates outperform the response with the first policy due to the additional operating rooms in the system.

Our analysis strategically locates the additional operating rooms in the periphery, mainly in the southernmost and northernmost zones (Fig. 6d). By deploying field hospitals in the periphery, more patients can be treated there, offloading the hospitals in the center. With this strategy, 2.1k people, representing 45% of the total patients, are treated in the four previously highlighted centric districts, 15% fewer patients than with the third strategy. As a result, fewer patients have to be transported from the periphery to the city center, offloading critical roads even more. In this case, the southernmost hospital will only have to transfer 54 patients to the largest hospital, a 29% reduction than with the first policy. With this strategy, only 15 additional operating rooms were deployed, representing only a 17% increase with respect to the mean number of functioning operating rooms in the entire city after the earthquake. If the EMTs deploy more operating rooms using this methodology, then treatment times will be further reduced, the periphery will be better supplied with needed resources, and the usage of ambulances and critical roads will be further offloaded in the city.

## Discussion

We present a methodology for characterizing the emergency response of hospital systems after earthquakes and designing policies to treat patients effectively. Our methodology establishes the groundwork for assessing the value of hospital system coordination through a metric that measures the performance of high-coordination emergency policies in terms of waiting times and effective use of hospital resources. Because our methodology considers hospitals in a large urban center to behave as a system, emergency managers and resilience officers can apply our methodology to a whole large city and evaluate optimal patient transfer strategies between hospitals, effectively allocate ambulances in the system, and guide the deployment of field hospitals.

We found that a M 8.0 earthquake in Lima will cause an emergency response with 4.7k critically injured people requiring surgical procedures, which will impose large demands on a hospital system that will work only at 48% capacity owing to earthquake disruptions. As in previous emergency responses, triage procedures will be key to identifying these critically injured patients and to enable doctors to give them priority for treatment, e.g., 2004 Sumatra earthquake in Indonesia[34], 2011 Christchurch earthquake in New Zealand[35]. In addition, the activation of

reverse triage protocols in Lima can help offload demands by releasing minor wounded patients to focus on these critically injured patients as it happened in the 2005 Pakistan earthquake[36] or even in Hurricane Sandy[37].

We also found that the criticality of the emergency response will be exacerbated by a spatial distribution of casualties that does not match the post-earthquake capacities of the hospital system (Fig. 3). The zones with higher post-earthquake capacity are located in the city center, in clear contrast with the zones with higher post-earthquake demands of health services. Large numbers of patients are located in the periphery, where, unlike the city center, deficient construction practices have rapidly increased the seismic vulnerabilities of the housing infrastructure[38]. The neighborhoods in the periphery tend to be populated by families with less income and wealth, so the disparities in disaster risk overlap with the economic disparities[39]. This overlap will exacerbate the critical conditions in the periphery because these families will have less resources to obtain treatment and medicine from private hospitals, relying mostly on public hospitals. During an emergency response, this uneven vulnerability profile in the housing sector exacerbates the resource centralization problem of hospitals in the city, thus leaving the neighborhoods in the periphery predominately underserved during an emergency response. Because in many cities, neighborhoods in peripheral zones have precarious access to health services[40] and high concentration of seismic vulnerabilities[41], these observations in Lima can be extrapolated to multiple urban centers in Latin America and even in developed countries.

Emergency planners who aim to treat patients in the city effectively must address these disparities by either transporting patients from lower-capacity to higher capacity zones or supplying the lower-capacity zones with additional resources to meet demands of health services. Though emergency managers can elaborate multiple reasonable strategies to implement such emergency response measures, our findings shows that strategies based on deeper coordination between hospitals prove to be significantly more effective than the ones with less ability to coordinate (Fig. 5). We chose four strategies owing to their conceptual simplicity to represent two scenarios with low-coordination capacities and other two with high-coordination capacities. Although simple conceptually, these strategies involve quite complex specific protocols for each hospital that need to be adjusted according to the hospitals' resources and the number of remaining patients in triage areas. Nevertheless, we show that our proposed formulation can find effective protocols under the high levels of complexity of the earthquake emergency response in Lima.

The first two strategies require low-coordination capacities, and as a result, they will perform poorly because hospitals will respond mostly locally. With the first strategy, hospitals will not establish transfer protocols unless they lose all their operating rooms. If they lose them, hospital directors will need to quickly communicate with the closest hospital that can conduct surgical procedures and establish reliable coordination for patient transfers using equipment for radio or satellite cell phone communications. The first strategy distributes the number of people treated at each hospital roughly evenly (Fig. 6a). Such a treatment distribution results in hospitals with fewer resources treating similar patient numbers as hospitals with more resources, making the system inefficient and increasing waiting times.

With the second strategy, hospitals also respond mostly locally. If a hospital loses all its operating rooms, the director will have to quickly communicate with the hospitals with the largest number of functional operating rooms in the city to establish a patient transfer procedure using a similar communication protocol to the first strategy. Yet, contrary to the first strategy, the second one

distributes the number of treated patients highly unevenly (Fig. 6b). Because multiple hospitals often end up transferring patients to the two largest hospitals in the city, Hospital Edgardo Rebagliati and Hospital Guillermo Almenara, these two hospitals largely overflow their capacities. In addition, the second strategy requires heavy use of ambulance resources that can overflow road capacities when transferring patients from the periphery to the city center, where these two largest hospitals are located.

In contrast, the third and fourth strategies have stronger coordination capacities that significantly improve the emergency response performance because under such emergency protocols, hospitals can share resources and leverage strategic system-level information. Existing legal frameworks in Peru can facilitate the activation of these high-coordination strategies. Owing to the severity of this emergency, the government will very likely trigger a declaration of State of Emergency as it happened after the M 8.0 2007 Pisco earthquake to facilitate the entry of international humanitarian assistance and the rapid purchase and deployment of goods and services for the emergency response[28]. With these strategies, ambulances are shared across the system to be effectively used in critical zones, thus they do not bottleneck the system as in first baseline strategy. Patients are also strategically transferred across the entire hospital system, leading to a spatial distribution of treated people that matches the post-earthquake capacities of the hospital system (Fig. 6c). As a result, the city-wide-average waiting times to treat patients decrease significantly by factors larger than 2 when compared with the previous baseline strategies that only allowed reduced pairwise coordination (Fig. 5).

Emergency response agencies under MINSA and Essalud, Servicio de Atención Móvil de Urgencia (SAMU), and Sistema de Transporte Asistido de Emergencia (STAE), respectively, have capacities to coordinate across multiple hospitals on a system level. SAMU and STAE must ensure the continuity of their existing communication capacities and enhance their coordination capacities to be able to direct the hospitals' ambulances strategically in Lima during an emergency response. Acquisition of hardware such as satellite cell phones and radios for all ambulances and hospitals and implementation of robust software to manage and process the patient treatment, triaging, and transfer information will be key to accomplishing such levels of coordination. Past earthquakes have shown that emergency responses with high coordination are feasible even with older technology. After the 1994 Northridge earthquake, the Los Angeles County Emergency Medical Services Agency coordinated with 76 hospitals using radio communications. The agency assessed the functionality of hospitals, evaluated which healthcare facilities were overloaded, and coordinated patient mobilization at the county level effectively[42].

To deploy these high-coordination strategies during the emergency response, SAMU and STAE will need a few days after the next large earthquake to collect necessary system-level information including both casualties and the residual hospital functionality[43,44]. They could use the trends shown in Fig. 6c for earthquake preparedness to properly plan the pragmatic implementation of effective strategies for the next earthquake emergency responses in Lima. These plans may include implementing reliable communication lines between hospitals more likely to transfer large number of patients. In addition, preparedness plans may include a usage prioritization of critical emergency corridors that are more likely to be heavily used by ambulances or an identification of alternative roads in case these corridors have known seismic vulnerabilities.

Finally, our findings also show how EMTs can strategically deploy mobile operating rooms leveraging high-coordination capacities and system-level information. In past earthquakes, EMTs have been key to deploying such life-saving resources worldwide[20], e.g., 1999 Turkey earthquake[45], 2004 Sumatra earthquake[46]. Using similar hardware and software requirements for system-level coordination to the third strategy, the fourth strategy can inform where and how many of the 15 mobile operating rooms should be deployed by EMTs, further demonstrating that by supplying the periphery with additional medical resources, the city will be more successful at decreasing global patient waiting times and offloading demands for patient mobilization. Without coordination, EMTs might find it practical to locate these additional operating rooms at the city center, where equipment mobilization is easier, and the availability of doctors and nurses is higher. However, such a plan will lessen the ability of the hospital system to effectively treat patients because more patients would have to be transferred from the periphery to the center, overloading the roads and potentially overflowing ambulance capacities.

In large urban centers, robust earthquake preparedness plans should be developed based on a thorough understanding of the uneven spatial distribution of capacity and demand of health services in an earthquake aftermath. Though this paper provides novel understanding on the spatial mismatch between capacity and demand and describes strategies to address it, more research is needed to understand how the vulnerability of other critical urban systems, such as the transportation or power system, play a role during the emergency response of the city. Robust methods for capturing the role of these urban systems have already been developed[47]; however, lack of exhaustive data sets with exposure and vulnerability information make such analyses prohibitive in most cities. Our emergency response assessment can help direct data collection efforts to hotspots in the city, such as to specific roads that are critical for emergency response, or specific areas where the city must ensure that the power supply is sustained or rapidly recovered. Effective plans will capitalize on the methodology and information provided here to better prepare cities facing high significant risk from future large earthquakes.

## Methods

**Network flow model and optimization formulation.** We model the post-earthquake hospital treatment process as a minimum cost time-varying network flow (MCTVNF) problem[48,49]. In our MCTVNF formulation, a directed graph $\mathbf{G} = (\mathbf{N}, \mathbf{E})$ represents the hospital system, where $n = |\mathbf{N}|$ is the number of graph nodes, and $e = |\mathbf{E}|$ is the number of graph edges. We use a discrete time model with a finite time horizon $t_f$ with time-steps $dt$, thus the time $t \in \mathbf{T}$: $\{0, dt, 2dt, …, t_f\}$. At each time $t$, each hospital has two nodes: one triage node where patients are received into the hospitals, and one discharge node where patients go after they complete their treatment. Each graph node is associated to an index $i$ and a time $t$, where hospitals' triage areas have indexes $i \in \mathbf{\Gamma}$: $\{1, 2, …, n_h\}$, and the discharge areas have indexes $i \in \mathbf{\Lambda}$: $\{n_h + 1, n_h + 2, …, 2n_h\}$, where $n_h$ is the number of hospitals in the system. To define a one-to-one correspondence in the indices of a hospital, if its triage index is $i \in \mathbf{\Gamma}$, then its discharge index is $i + n_h \in \mathbf{\Lambda}$. Fig. 7 shows an example of a network representation at time $t$ for a system with three hospitals, where the triage nodes are in red and the discharge nodes in blue. In this model, the decision variables are both the flows through the edges and the patient queues in thee triage nodes. These variables will track how many patients will stay in triage, be treated or be transferred to other hospitals.

Each graph node is associated to a time-variant demand-supply variable $b_i(t)$. In triage nodes, $b_i(t)$ represents the number of people arriving to hospital, thus they are analyzed as source nodes with non-negative flows: $b_i(t) >= 0, \forall i \in \mathbf{\Gamma}$. In the discharge nodes, $b_i(t)$ represent the number of patients who finish their treatment and exit the hospital at time $t$, thus they are analyzed as sink nodes with non-positive flows: $b_i(t) <= 0, \forall i \in \mathbf{\Lambda}$. We assume that patients who finish the treatment process and exit the hospital do not to return to the hospital system during the time horizon $t_f$. Each graph edge is associated to a flow of patients $x_{i,j}(t)$ that leaves node $i$ at time $t$ to go to node $j$. In this formulation, edges fully connect the triage nodes to allow hospitals to redistribute their patient loads to potentially any other hospital within the system according to their available ambulances. In addition, each triage node is connected to its respective discharge node to represent the patient treatment process within a hospital. Fig. 7 shows the edges and respective flows between triage nodes from different hospitals and between triage and discharge nodes within same hospitals for the system with three hospitals. At each time $t$, the flow $x_{i,j}(t)$ has a maximum bound $u_{i,j}(t)$ and a travel time $\tau_{i,j}(t)$. In

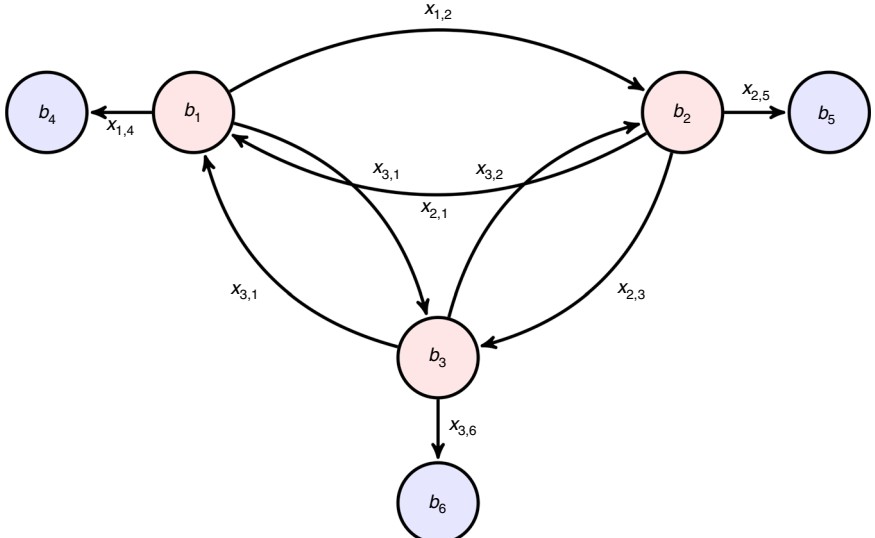

**Fig. 7 System model with three hospitals at time _t_ as a directed graph.** The system model used for the application to Lima has 41 hospitals, i.e., 41 triage and 41 discharge nodes.

this discrete formulation, it is considered that the flow $x_{i,j}(t)$ leaves the node $i$ at time $t$ and reaches the node $j$ at time $t + \tau_{i,j}(t)$.

For the edges connecting triage areas, $u_{i,j}(t)$ represents the maximum number of patients who can be transported from triage $i$ to triage $j$ in a different hospital according to the available transportation resources (e.g., ambulances available in the hospital), and $\tau_{i,j}(t)$ is the transportation time of the patients from triage $i$ to triage $j$. For the application to Lima, $u_{i,j}(t)$ and $\tau_{i,j}(t)$ in these edges were defined according to the ambulance capacities in each hospital and the travel times from pre-earthquake traffic conditions, respectively. When vulnerability data for the transportation system in Lima is available, our model will be able to leverage existing risk models for transportation systems[50] to adjust travel times to post-earthquake traffic conditions.

For the edges connecting triage nodes $i$ with their respective discharge nodes $j = i + n_h$, $u_{i,i+n_h}(t)$ represents the maximum number of patients who can be treated according to the available medical resources for the type and severity of the patients' injuries, and $\tau_{i,i+n_h}(t)$ is the treatment time. For the application to Lima, $u_{i,j}(t)$ and $\tau_{i,j}(t)$ in these edges were defined according to the functional operating rooms in each hospital and average treatment times in operating rooms in previous earthquakes[51].

In addition, we define $y_i(t)$ as a storage variable at each triage node to represent the patients who wait in the hospital queue to either be treated within the hospital or be transported to another hospital with more available resources.

**Optimization of performance metric.** We evaluate both waiting times and effective use of ambulances as the system performance metric, thus the metric includes two objective functions. The first objective function $C_1(X)$ measures waiting time across the city as the average time that a patient would take since the earthquake until completing treatment in the operating room.

$$C_1(\mathbf{X}) = \frac{\sum_{t \in \mathbf{T}} \sum_{\substack{i \in \boldsymbol{\Gamma}, \\ j = i + n_h}} \{t + \tau_{i,j}\} \times x_{i,j}(t) \times dt}{\sum_{t \in \mathbf{T}} \sum_{i \in \boldsymbol{\Gamma}} b_i(t)} \quad (1)$$

$\mathbf{X}$ represents a vector containing all the decision variables of flow $x_{i,j}(t)$ in edges and the storage $y_i(t)$ in the triage nodes. The numerator of $C_1(\mathbf{X})$ represents the total number of patients passing through the operating rooms (from each triage node, $i \in \boldsymbol{\Gamma}$ to the corresponding discharge node $j = i + n_h \in \boldsymbol{\Lambda}$) multiplied by their respective times to complete treatment, whereas the denominator is the total number of patients arriving to the triage areas. The time horizon $t_f$ is carefully chosen to have enough modeling time to treat the patients. However, in few simulations with significant number of patients and not many functional operating rooms, a couple of terms are added to the numerator, one with the remainder patients in the triage areas, $t_f \times \sum_{i \in \boldsymbol{\Gamma}} y_i(t_f) \times dt$, and another with the remainder patients in the ambulances, $t_f \times \sum_{i \in \boldsymbol{\Gamma}} x_{i,i+n_h}(t_f) \times dt$, in order to properly incorporate the unmet demands at the end of the simulation into $C_1(\mathbf{X})$.

The second objective function measures ambulance usage as the total number of patients transported in ambulances. The objective function $C_2(\mathbf{X})$ is normalized by

the total number of patients analogously to $C_1(\mathbf{X})$.

$$C_2(\mathbf{X}) = \frac{\sum_{t \in \mathbf{T}} \sum_{i \in \boldsymbol{\Gamma}} \sum_{j \in \boldsymbol{\Gamma}} x_{i,j}(t) \times dt}{\sum_{t \in \mathbf{T}} \sum_{i \in \boldsymbol{\Gamma}} b_i(t)} \quad (2)$$

We define a system cost $C(\mathbf{X})$ as a weighted sum of $C_1(\mathbf{X})$ and $C_2(\mathbf{X})$ to find a Pareto-optimal solution.

$$C(\mathbf{X}) = \alpha_1 \times C_1(\mathbf{X}) + \alpha_2 \times C_2(\mathbf{X}) \quad (3)$$

After assessing multiple $\alpha_1$ and $\alpha_2$ values, we minimized $C(\mathbf{X})$ using values of 0.90 and 0.1, respectively. Smaller $\alpha_2$ values resulted in inefficient ambulance usage with small reductions in waiting times, requiring some patients to be transferred multiple times in ambulances before being treated. Larger $\alpha_2$ significantly increased waiting times, thus these $\alpha_2$ values do not appropriately represent that the priority in the formulation is to minimize waiting times over to use ambulances with efficiency. We find the best set of decisions $\hat{\mathbf{X}}$, vector that contains the values of flow variables $x_{i,j}(t)$ and storage variables $y_i(t)$ which minimize $C(\mathbf{X})$.

$$\hat{\mathbf{X}} = \operatorname{argmin}_{x_{i,j}(t); y_i(t)} \quad C(\mathbf{X}) \quad (4)$$

The decision variables are subject to the constraints in Equations (5), (6), (7), and (8). Equation (5) represents patient flow conservation, which guarantees that all the patients coming into the hospital system stay within the system until they leave through the discharge nodes.

$$x_{i,i+n_h} + \sum_{j \in \boldsymbol{\Gamma}} x_{i,j}(t) - \sum_{j \in \boldsymbol{\Gamma}} x_{j,i}(t - \tau_{i,j}(t)) + y_i(t + dt) - y_i(t) = b(i), \quad \forall i \in \boldsymbol{\Gamma}, t \in \mathbf{T} \quad (5)$$

Equations (6) and (7) represent flow capacity constraints. Equation (6) ensures that the people in the operating rooms do not exceed the unitary capacities $u_{i,i+n_h}$, where $u_{i,i+n_h}$ is estimated as the number of functional operating rooms in the hospital $i$ over the number of surgeries per day. We assumed that each surgery takes 4 hours, and that hospitals will be functional 24 hours during the emergency response using multiple personnel shifts. Such treatment rate equals the rates in foreign field hospitals after the 2004 Indonesia earthquake/tsunami[51].

$$0 \le \frac{x_{i,i+n_h}(t)}{u_{i,i+n_h}} \le 1, \quad \forall i \in \boldsymbol{\Gamma}, t \in \mathbf{T} \quad (6)$$

Equation (7) ensures that the patient transfers do not exceed the total unitary transportation capacities in a hospital, where $u_{i,j}$ is the unitary capacity if all ambulances of a hospital were only transferring patients from triage $i$ to $j$. $u_{i,j}$ equals the number of ambulances in the hospital times the number of patients transported per ambulance trip over the number of round trips that an ambulance can make from triage node $i$ to $j$. We retrieved travel time information from Google Maps API to estimate the round trip numbers and assumed that each ambulance trip can take up to two patients.

$$0 \le \sum_{j \in \boldsymbol{\Gamma}} \frac{x_{i,j}(t)}{u_{i,j}} \le 1, \quad \forall i \in \boldsymbol{\Gamma}, t \in \mathbf{T} \quad (7)$$

Equation (8) ensures that the number of patients waiting in the hospitals' triage queues are properly represented by a non-negative number.

$$0 \leq y_i(t), \quad \forall i \in \Gamma, t \in \mathbf{T} \tag{8}$$

Equations (6), (7), and (8) introduce a model relaxation. Whereas the number of patients who are treated, transported or waiting in the queue can only be non-negative integers, the formulation expands the variables' domain to include real numbers. This relaxation ensures that the formulation is tractable. Thus, because the cost and the constraint functions are linear combinations of the decision variables, we solve this minimization as a linear programming problem using the simplex algorithm in GLPK of the cvxopt implementation in Python[52].

**Model adaptation for baseline strategies 1 and 2.** Both baseline strategies have limited coordination capacity and only allow each hospital to transfer patients to only one single hospital with functional operating rooms instead of multiple ones. Thus, to represent these strategies, the model ignores multiple transfer edges in the flow model, reducing the elements of the edge set **E**. In the first baseline strategy, only the edges going from hospitals without functional operating rooms to the closest hospitals are activated. In the second baseline strategy, only the edges going from hospitals without functional operating rooms to the hospital with the largest number of functional operating rooms are activated.

Because the model is solved multiple times according to the number of patients and functional operating rooms in the earthquake simulation, then the edge connectivity varies from simulation to simulation. With strategies 1 and 2, the number of edges in the model is significantly reduced, thus we modeled larger time horizons. We selected a time horizon $t_f$ of 100 days, which is sufficiently long period to treat all earthquake patients in most simulations, and a time step $dt$ of 1 day.

**Model adaptation for sharing ambulances.** Strategy 3 does not need to disconnect edges in the model. Yet, it modifies the transportation edges' capacity constraints to enable hospitals to share ambulance resources. Thus, the constraint in Equation (7) is relaxed as follows.

$$0 \leq \sum_{i \in \Gamma} a_i \sum_{j \in \Gamma} \frac{x_{i,j}(t)}{p_{i,j}} \leq \sum_{i \in \Gamma} a_i, \quad \forall t \in \mathbf{T} \tag{9}$$

Equation (9) ensures that unitary transportation capacities are not exceeded at a system level at each time step, where $a_i$ represents the number of ambulances of hospital $i$. All the other constraints remain the same. Because modeling this policy requires higher edge connectivity than the baseline strategies and thus has more computational demands, the time horizon $t_f$ was reduced to 40 days. It was verified that such a variation did not affect the optimization because less modeling time was needed as a result of shorter optimal waiting times with the strategies 3 and 4 (Fig. 5). The time step $dt$ was kept equal to 1 day.

**Model adaptation for deployment of more operating rooms.** Strategy 4 requires an additional modification to the constraint on the operating room capacity in Equation (6). This strategy allows EMTS to increase hospital capacities by introducing additional mobile operating rooms in close proximity to them as follows.

$$0 \leq \frac{x_{i,j}(t) - q_i}{u_i} \leq 1, \quad \forall i \in \Gamma, j = i + n_h \in \Lambda, t \in \mathbf{T} - \{0, dt, \ldots, t_s\} \tag{10}$$

Equation (10) ensures that hospitals can increase their unitary operation room capacities by $q_i$ after the time $t_s$ at which the operating rooms in the field hospitals are deployed in the city. In addition, the sum of the additional resources distributed across the system cannot exceed the total capacity $Q$ supplied by all the field hospitals in the region as follows.

$$0 \leq \sum_{i \in \Gamma} q_i \leq Q \tag{11}$$

All the other constraints remain the same. These modifications barely change the optimization complexity. Thus, we kept the time horizon equal to 40 days and the time step equal to 1 day.

**Earthquake casualty modeling.** We utilize an earthquake multiseverity casualty model previously developed by the authors[17] to evaluate the spatial distribution of injuries requiring surgical treatment after the M 8.0 earthquake. The model is probabilistic and uses ground shaking estimates to propagate the earthquake intensity to building damage according to the building seismic vulnerability[53] and the site-specific soil conditions in Lima[54]. Next, the model uses information on building occupancy to provide probabilistic estimates of the spatial distribution of injuries and fatalities in the city. The validity of the model results was verified[16] by comparing the casualties and fatality levels in the city to empirical formulas[55] and with fatality-to-collapse building data from the 2005 Pakistan earthquake[56].

The model categorizes injuries into three severities. The second- and third-degree severity require specialized medical attention and hospitalization, however, unlike the second degree, the third one requires immediate rescue and treatment to avoid death[57]. We considered that 100% of the patients with third-degree injuries, for example, having punctured organs or crush syndrome with exposed wounds,

plus 10% of patients with second-degree injuries, for example, having compound bone fractures, will require surgical treatment in operating rooms. We considered that patients arrive to the closest hospital during a period of 4 days after the earthquake in accordance to the evidence from previous earthquakes[32]. Thus, in the flow model the demand-supply variable $b_i(t)$ is larger than 0 in the triage nodes during the first four days after the earthquake. We considered that patients wait in triage zones to until an operating room is available in the hospital or until they are transferred to other hospitals.

**Seismic analysis for hospital functionality.** We utilize earthquake simulation to model the functionality of operating rooms during the emergency response[58]. Hospitals are complex infrastructure, whose post-earthquake functionality depends on multiple components: structural damage; damage in mechanical, electrical components and medical equipment; utility failure; shortage of medical supplies (i.e., oxygen, blood), and shortage of medical personnel[9,10,14,59]. Hospitals with slight structural damage can lose partial or total functionality as a result of damage and loss of the other components of hospitals[60].

To capture these effects, we analyzed that the structural vulnerability[53] of the +700 buildings belonging to the 41 healthcare campuses in the city according to the earthquake shaking intensity and the soil conditions on site. Then, we used a Bernoulli distribution to model loss of functionality that can occur owing to failure of components different to the hospitals' structure according the Hospital Safety Index (HSI). HSI is based on a qualitative evaluation of multiple hospital components including buildings' nonstructural elements such as equipment and backup medical resources, and technical and organization capacities in the hospitals' personnel[29]. HSI has three categories: A, B, and C, ranging from the best to the lowest performance. We used a different Bernoulli distribution for each HSI category. We considered that operating rooms in buildings with no structural damage have 1, 0.75, and 0.5 of functionality probability for categories A, B, and C, respectively, whereas that in buildings with slight structural damage, operating rooms have 0.6, 0.45, and 0.3 of functionality probability. Operating rooms in buildings with larger damage levels were considered completely non-functional.

The 41 campuses in the data set are part of the public healthcare system led by the Peruvian Health Ministry (MINSA) and the Social Security (Essalud). Even though there is a growing private healthcare system, most of the healthcare services are provided by the public system in Lima[61]. Physicians who work full time in the public healthcare system often work part-time in the private system[62], thus in an emergency, they would aim to provide services in the public system rather than in the private one. We consider that studying the response of the public sector represents a robust starting point to characterize the earthquake emergency response of the hospital system in Lima.

We supplemented the hospitals' building information with the number of ambulance in each hospitals. Because, a few hospitals have no ambulances, we considered that during the emergency response the local government or private institutions will supply one ambulance to each of these hospitals so that each hospital is able to mobilize patients.

**Earthquake shaking.** We studied the tectonics of the M 8.0 1940 earthquake and located the rupture area in the region delimited by the earthquake aftershock zone[23]. We defined the rupture dimensions along the fault strike and dip directions using an empirical function based on subduction zone earthquake data[63].

Next, we evaluated the ground shaking in a grid of 1 km × 1 km using site-specific lognormal distributions. We evaluated three ground shaking intensity measures, peak ground acceleration spectral acceleration at 0.3s, $Sa(0.3s)$, and spectral acceleration at 1s, $Sa(1.0s)$. We selected these intensity measures to better capture the response of multiple typologies of buildings in the inventory according to their predominant period of vibration. The log-mean and log-standard-deviation values of the intensity measures were extracted from empirical formulas that relate magnitude, site distance, and soil conditions to the ground shaking[64]. We included within-[65] and between-[66] event correlations in the intensity measures. The between-event correlations introduce spatial correlations to the ground shaking.

**Model limitations and future work.** Our formulation and case study advance the field of earthquake emergency response. However, there are existing limitations that must be addressed in future research. First, we model patient transfers that heavily rely on the proper functioning of the transportation system during the emergency response. However, our paper does not capture that the transportation network can be disrupted owing to potential bridge failures, seismic liquefaction, or infrastructure debris caused by the earthquake. Although there are models to capture these disruptions in a large city, we did not capture them because they require extensive seismic vulnerability data currently not available for Lima. Only a few large urban centers worldwide have sufficient vulnerability data to start capturing disruptions in the entire transportation network, e.g., the Bay Area in California[50]. Because collecting such data often requires significant resources from cities, strategic surveying of critical elements (e.g., critical bridges and roads) in the system can be an effective starting point for cities that lack sufficient vulnerability data. Our results can be used to support such a strategic surveying approach in large urban centers Instead of exhaustively collecting the vulnerability data of all components of the transportation network, the de facto approach in risk analysis,

researchers can use our results to prioritize surveying the critical roads for patient transfers to evaluate its potential disruptions owing to structural failure, soil liquefaction, and debris blockage and propose strategic risk mitigation measures for the network. Similarly, the power, water, and sewage networks[67] are pivotal for the proper functioning of the hospital facilitates. Our results can help future research focus on the power substations, power distribution lines, and water and sewage pipes serving the critical hospitals in the system. Focusing on these critical components will enable large urban centers which lack data to have a robust starting point for including the interactions between the hospital system and other critical urban systems.

Second, our case study only included an evaluation of an earthquake occurring at nighttime because we lacked data with the spatial distribution of the urban population at other times of the day. This spatial distribution is an important factor in the assessment of the spatial distributions of earthquake casualties because it enables researchers to track what vulnerable buildings are occupied. In our nighttime scenario, people are mostly at residential buildings, which are particularly vulnerable in Lima. We used LandScan information, which is available worldwide[68], as a proxy for the nighttime distribution of the population[16,17]. Earthquake scenarios occurring during the daytime will have a different distribution of population and therefore a different spatial distribution of earthquake casualties. Although we were not able to track the time variations of urban population in Lima, existing work has already shown that mobile network data can be effectively used for it, e.g., Beijing in China[69]. As more mobility data are rapidly collected by public and private institutions, plenty of opportunities to create large databases with time-variant high-resolution urban densities will be available in the near future. Deployments of models to track these time-variant urban densities will enable researchers to leverage the formulation presented here and extend the emergency response application to different times of the day in multiple cities so that emergency managers are better informed on how to protect their communities.

**Reporting summary**. Further information on research design is available in the Nature Research Reporting Summary linked to this article.

## Data availability

All the data to reproduce the findings of the paper can be found at https://purl.stanford.edu/dp530wq8437. Source data for the figures presented in the paper are also provided in the same link.

## Code availability

All the computer code to reproduce the findings of the paper can also be found at https://purl.stanford.edu/dp530wq8437.

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

## Acknowledgements

We thank Dr. Maryia Markhvida, Abhinav Bindal, and Jacqueline Li for helping conceptualize the first version of the patient transfer analysis during an emergency response. We also thank professor Sandra Santa Cruz, from the Pontificia Universidad Católica del Perú for granting access to the hospitals' seismic vulnerability information, and professor Carlos Zavala and Miguel Estrada from the Centro Peruano-Japonés de Investigaciones Sísmicas y Mitigación de Desastres (CISMID) and the Universidad Nacional de Ingeniería (UNI) for providing access to the seismic microzonation data in Lima. We thank Dr. Ken Snyder, Dr. Juan Fung, and Dr. Siamak Sattar from NIST for providing valuable feedback for our paper and Jill O'Nan from Stanford University for helpful revision of the paper writing. We acknowledge the financial support by the John A. Blume Fellowship from the Civil Engineering Department at Stanford University. In addition, this research was partially supported by the NSF Grant 1645335. The authors are grateful for their generous support.

## Author contributions

L.C., A.K., and G.D. conceived the basic emergency response network model, curated the data to model post-earthquake hospital functionality and set up the earthquake casualty model. L.C., J.M., A.K, and G.D. reviewed and refined the hospital functionality model to focus on high-severity injuries and defined the metrics to measure the effectiveness of emergency response plans. L.C., J.M., and C.B. defined the strategies for emergency response included in the paper. C.B. provided further guidance on the modeling of the deployment of EMTs and ambulance sharing. L.C. drafted the manuscript with contributions and editing from all the authors.

## Competing interests

The authors declare no competing interests.
