## [Peer Review File · Nature Communications]

Reviewers' comments:

Reviewer #1 (Remarks to the Author):

The manuscript is a welcome contribution in the realms of evaluating post-earthquake performance of hospital system in Lima, Peru. This is a relatively under-examined research area in general despite the fact that the hospitals play a critical role in saving enormous number of lives, serving to the injured population, and enabling speedy recovery. I thank the authors for undertaking this research by looking at the hospital system in its entirety instead of individual hospital response. The proposed approach is novel, the loss estimation framework is robust, and the results are worthy of a publication.

I do have few minor comments which I hope can be incorporated in the revised submission:

1. The manuscript does need some work in terms of writing. There are number of typos (e.g., ETMs), grammar issue, and repetitiveness of certain points (e.g., four strategies are discussed multiple times).
2. The manuscript can benefit from some reorganization. Certain points are introduced too briefly without any explanation (e.g., network flow model, incorporating traffic data from google API) and the overall flow of information does not seem to make sense in the first reading (results, their impact are discussed ahead of the discussion on causal factors, method and analysis).
3. The biggest weakness of the manuscript is that it completely lacked discussing: (a) key data on hospital responses from previous earthquake (within Peru and elsewhere, e.g., Japan, Taiwan, Turkey, Chile, Indonesia), (b) how hospitals currently plan and respond to large earthquake or other disasters (state-of-the-practice), (c) how practical it is to assume that the key hospital resources (such as ambulance) can be shared freely across multiple cities OR deployment of additional EMTs and the availability of resources (e.g., space, medical professionals) when main facilities are already overwhelmed and crowded. Some discussion is needed on the state-of-practice, placement protocols, etc.
4. The mean and 90th percentile estimate of city-wide waiting times between strategy 3 & 4 are remarkable close (10-19 & 8-15 days). It is not entirely clear whether in strategy 4 authors assumed ambulance sharing option was available or not (line 249?). The authors need to tabulate all the results clearly.
5. Discussion section need a lot of work. The authors need to emphasize on significance of each strategy, their practicability, significance of results, alternate ways for improvements in enhancing capacity, limitations of proposed method, practical constraints, and scope of future research more clearly.

Reviewer #2 (Remarks to the Author):

The paper "Effective Plans for Hospital System Response to Earthquake Emergencies" is an interesting work, but I don't think that it's appropriate for Nature Communications publication. I don't see what the new discovery or significant result of general interest is that would make this appropriate for Nature Communications. I'm sure there are more specialized journals that this would be appropriate for.

The authors say that "a methodology to evaluate emergency response" is presented and that this proposed methodology "is able to design effective plans for patient transferal and allocation of medical resources". In the paper though this methodology is only used to evaluate four very simplistic emergency response plans the outcomes of which are obvious. In order to apply this methodology the "unique" dataset of high-resolution hospital vulnerability is needed and that might not be generally available. Also, the proposed metric is based on patient waiting times for the emergency room use and effective use of ambulance patient transfers, but there are other important parameters, like the number of patients a hospital can hold in general. One more shortcoming of this study is that the authors consider only a nighttime scenario, which significantly reduces the generality of the approach. During the daytime, many people who live in the

peripheral regions might be in the center of the city.

It appears that a motivation of the paper is that cities are encouraged by relevant authorities to reach some standard for being prepared for providing medical treatment after an earthquake, and cities can follow the approach described in the paper to compare the quality of the medical care they could provide if they follow each of the scenarios described. Presumably one expects the more complex scenarios to be better, but perhaps the question is whether the added cost is worth the improvement in medical care or patient outcome. The paper should really have much more discussion of how to apply the approach to arbitrary cities, and maybe calculate the cost of the different scenarios to aid decision making.

There are a number of grammar mistakes and typos in the paper, for example:

line 85: ETMs instead of EMTs

line 99: extra "to" between "effectively" and "treat"

line 439: "the an" at the end of the line

line 458: extra "are"

line 543: "distributions" is missing the "b"

line 463: "treat" is repeated twice

line 568: "the" is repeated twice

The figures are out of order, for example Figure 9 is mentioned on line 122 before Figure 4. The subplots a and b in Figure 5 should be combined for a better visibility and easy comparison.

The formulation itself of the network flow model is promising but I don't think that it's used to its full potential and that the whole methodology can be widely applied and thus I do not believe it is worth considering for publication at its current stage.

The link for the data and the code did not work for me.

Reviewer #3 (Remarks to the Author):

1-Please highlight the novelty of your research. The differences of your work with other researches in this field is not clear.

2-Please place the "Acknowledgment" section at the end of manuscript (before reference section).

3- Some new published relevant articles such as:

Capacity planning and reconfiguration for disaster-resilient health infrastructure by

Amir Hossein Aghapour et al, Journal of Building Engineering

Volume 26, November 2019.

should be included in the manuscript.

I believe the revised manuscript by considering the above comments could be accepted for publication.

Rebuttal Letter for Nature Communications

Reviewer #1 (Remarks to the Author):

The manuscript is a welcome contribution in the realms of evaluating post-earthquake performance of hospital system in Lima, Peru. This is a relatively under-examined research area in general despite the fact that the hospitals play a critical role in saving enormous number of lives, serving to the injured population, and enabling speedy recovery. I thank the authors for undertaking this research by looking at the hospital system in its entirety instead of individual hospital response. The proposed approach is novel, the loss estimation framework is robust, and the results are worthy of a publication.

Dear reviewer #1,

Thank you for all your feedback. We agree with you that emergency response is critical for serving injuries, saving lives, and speeding the recovery process. We also agree with you that the topic has not been properly examined by the academic community. We appreciate all your suggestions and insightful comments and have updated our paper accordingly. Please, find our responses to your specific comments below.

I do have a few minor comments which I hope can be incorporated in the revised submission:

Comment #1: The manuscript does need some work in terms of writing. There are number of typos (e.g., ETMs), grammar issue, and repetitiveness of certain points (e.g., four strategies are discussed multiple times).

Response #1:

We have corrected all the typos and addressed grammar issues. We also modified the text to avoid repetitions of the term 'four strategies/plans' in lines 106, 198, and the caption of Figure 6. Additionally, we simplified the descriptions of the four strategies in the Introduction (line 92) so that the descriptions introduced in subsections 1.4, 1.5, and 1.6 are not redundant.

Comment #2: The manuscript can benefit from some reorganization. Certain points are introduced too briefly without any explanation (e.g., network flow model, incorporating traffic data from google API) and the overall flow of information does not seem to make sense in the first reading (results, their impact are discussed ahead of the discussion on causal factors, method and analysis).

Response #2:

We edited the paper as suggested. Because "network flow model" is a term that we use repetitively, we now introduce a brief explanation early on. Please, see line 64:

The network flow model captures the system behavior of emergency responses through the evaluation of patient treatment, triage process, and patient transfers across a large number of hospitals, which establishes a fundamental difference from existing formulations focused on analyses on a single-hospital scale.

Additionally, we described the specific data that was retrieved from the Google API (i.e., travel times) in section “Demand-capacity Mismatch of Health Services” without overloading the reader with information about the Google API. However, in the “Methods” section, we kept the explanation on the Google API so that the interested reader can locate this information. Please, see the update in line 187:

Combining simulations on post-earthquake demand-capacity with data including the available ambulances at each campus and the driving times between hospitals, we conducted a probabilistic evaluation of the system metric performance for the emergency plans.

And line 499 in the “Methods” Section:

We retrieved travel time information from Google Maps API to estimate the round trip numbers ...

Moreover, the format of Nature Communications asks us to show the “Results” section before the “Methods” section so that the reader can see the contributions of the paper without going through dense descriptions of the formulation. We followed this format. However, we agree with you that the paper needed more explanation on the causal factors of emergency response before introducing the “Results” section. We added more descriptions on the causal factors as well as on data from previous earthquakes in the “Introduction” as also suggested in your Comment #3. For example, see the paragraph in line 34:

Natural disasters also demand large mobilizations of patients. For example, 2k to 3k patients had to be transferred between hospitals after the M 8.8 2010 Chile earthquake [4]. In large urban centers, such massive mobilization of patients requires an organized and system-level response to relocate patients effectively and treat them timely. Yet, evidence shows that responses are often not the result of a global, system-level strategy but rather local and haphazard [5]. After the M 6.7 1994 Northridge earthquake, two hospitals transferred their patients to a hospital that was condemned itself, thus the patients had to be transferred a second time [6,7].

Additional discussions and supporting information related to other causal factors and characteristics of an emergency response model are added as part of the responses to your Comment #3.

Comment #3: The biggest weakness of the manuscript is that it completely lacked discussing:

Response #3:

Thank you for pointing this out. We strengthen the paper by adding the following discussions and information as follows:

Comment #3a: key data on hospital responses from previous earthquake (within Peru and elsewhere, e.g., Japan, Taiwan, Turkey, Chile, Indonesia),

Response #3a:

We extended the literature review to highlight data and information from previous earthquakes in the “Results” and “Discussion” section. The data supports the emergency response model as follows:

See line 252:

We assume that the additional operating rooms will be functioning three days after the earthquake. Though three days can be a short time to deploy additional operating rooms, well-

organized surgical teams have demonstrated that they can deploy their medical resources even within 48 hours after the disaster even from abroad, e.g., 2007 Pisco earthquake in Peru [46], 1999 Turkey earthquake [47]. This strategy deploys the operating rooms close to existing hospitals to leverage their triage areas and additional resources such as personnel, power generators or backup water. Such arrangements worked well during the emergency response following the 2010 Chile Earthquake and extended the existing capacity of healthcare facilities [4].

Also, see line 138:

We estimate that on average only 87 of 182 total hospital operating rooms (48%) will be functional after the M 8.0 earthquake. We verified that our results were consistent with the reductions of hospital capacities in past earthquakes. For example, similar to our predicted high levels of capacity reduction, the M 8.0 2007 Pisco earthquake in Peru reduced the hospital bed capacity to 38% [42].

Also, see line 42 (from Response to Comment #2):

Natural disasters also demand large mobilizations of patients from affected regions to the hospitals or through transfers between hospitals. For example, 2k to 3k patients had to be transferred between hospitals after the M 8.8 2010 Chile earthquake [4]. In large urban centers, such massive mobilization of patients requires an organized and system-level response to relocate patients effectively and treat them timely. Yet, evidence shows that responses are often not the result of a global, system-level strategy but rather local and haphazard [5]. After the M 6.7 1994 Northridge earthquake, two hospitals transferred their patients to a hospital that was condemned itself, thus the patients had to be transferred a second time [6,7].

These additional discussions and descriptions of past emergency responses complement the descriptions we already had in the initial version of the paper. For example, see line 28:

Natural disasters such as earthquakes, landslides, floods, typhoons put heavy demands on hospital systems because these disasters can cause thousands or even tens of thousands of injuries in a short timespan (Figure 1). At the same time, natural disasters cause massive disruptions to hospital systems by damaging their supporting infrastructure. For example, the M 7.6 1999 Turkey earthquake caused around 50,000 injuries in Izmit and disrupted 10 major hospitals, which required the relocation of most patients from these hospitals [3].

Or line 551:

The validity of the model results was verified [19] by comparing the casualties and fatality levels in the city to empirical formulas [23] and with fatality-to-collapse building data from the 2005 Pakistan earthquake [76].

Comment #3b: how hospitals currently plan and respond to large earthquake or other disasters (state-of-the-practice),

Response #3b:

We agree with you that the paper needed more discussion about the state-of-practice for earthquake emergency responses.

Countries and cities have existing legal frameworks that are activated during the response to an emergency. For example, after confirming the severity of the 2007 Pisco earthquake, the Peruvian Government declared the State of Emergency in the affected region [Bambaren, 2010]. The state of emergency allows the government to access emergency funds and speed up procedures to purchase goods and services so that they can be deployed quickly to address the needs of the affected people. The state of emergency also facilitates the entry of humanitarian dispatches that include medical supplies and teams to be deployed in the affected regions. We added the following paragraph in line 350 to point out to the potential “State of Emergency” in the region:

Due to the severity of this emergency, the government will very likely trigger a declaration of “State of Emergency” as it happened after the M 8.0 2007 Pisco earthquake to facilitate the entry of international humanitarian assistance and the rapid purchase and deployment of goods and services for the emergency response [42].

Though the legal framework for the state of emergency is well established in Peru, actual emergency responses are complex, uncertain, haphazard, and with multiple situations not anticipated or prescribed in the legal frameworks for emergencies. Similar issues also happened in past earthquakes in other countries (e.g., Tohoku earthquake in Japan in 2011 [Parmar et al., 2013]). We added the following example to highlight how haphazard can be emergency responses (same as the response to Comment #2):

Natural disasters also demand large mobilizations of patients from affected regions to the hospitals or through transfers between hospitals. For example, 2k to 3k patients had to be transferred between hospitals after the M 8.8 2010 Chile earthquake [4]. In large urban centers, such massive mobilization of patients requires an organized and system-level response to relocate patients effectively and treat them timely. Yet, evidence shows that responses are often not the result of a global, system-level strategy but rather local and haphazard [5]. After the M 6.7 1994 Northridge earthquake, two hospitals transferred their patients to a hospital that was condemned itself, thus the patients had to be transferred a second time [6,7].

However, there are specific repetitive patterns of emergency responses after earthquakes across multiple countries. The first two critical patterns are the increase in casualties and the reduction in hospital functionality. The initial version of the paper highlighted this issue referencing past examples of surges in casualties (for example, see Figure 1 and section 1.1) and also the reduction in hospital capacity (see section 1.2). The added discussions in the response to your comment 3a also address this point. An added critical pattern is the need to mobilize large numbers of patients either as part of transfers or evacuation procedures. This point was described in line 34 (same as the previous paragraph in red).

Another repetitive pattern in earthquake emergency responses is the triage procedure. Often patients are triaged into a few classifications according to the injury severity and the required medical treatment. For example, after the 2008 Wenchuan earthquake in China, the 2283 earthquake patients at the West China Hospital of Sichuan University, the largest hospitals of Sichuan province, were triaged into four main groups: resuscitation, urgent treatment, delayed treatment, and minor injuries (Nie et al., 2011). Similar triaging procedures with a few categories also occurred in other countries, e.g., in Indonesia after the 2004 Sumatra earthquake (Li and Zheng, 2014), in New Zealand after the 2011 Christchurch earthquake (Ardagh et al., 2012).

Also, as part of the emergency response, hospitals apply the principle of reverse triage (Kelen et al., 2009). Under a reverse triage protocol, hospitals offload the facility of the minorly wounded people and pay more attention to the seriously injured patients. E.g., 2005 Pakistan earthquake (Awais, 2013) or even

after hurricanes in Sandy (Wang et al., 2017). Both, triaging patients according to their injury severity and applying reverse triage procedures enable hospitals to focus on the critically injured patients. Please, see paragraph in line 291, where we expanded the discussion with these two points:

We found that a M 8.0 earthquake in Lima will cause an emergency response with 4.7k critically injured people requiring surgical procedures, which will impose large demands on a hospital system that will work only at 51% capacity due to earthquake disruptions. As in previous emergency responses, triage procedures will be key to identifying these critically injured patients and to enable doctors to give them priority for treatment, e.g., 2004 Sumatra earthquake in Indonesia [48], 2008 Wenchuan earthquake in China [49], 2011 Christchurch earthquake in New Zealand [50]. Additionally, the activation of “reverse triage” protocols in Lima can help offload demands by releasing minor wounded patients to focus on these critically injured patients as it happened in the 2005 Pakistan earthquake [51] or even in Hurricane Sandy [52].

References:

- Bambarén, C. (2010). Legal issues of humanitarian assistance after the 2007 earthquake in Pisco, Peru. *Prehospital and Disaster Medicine*, 25(3), 203–206. <https://doi.org/10.1017/S1049023X00008013>
- Parmar, P., Arij, M., & Kayden, S. (2013). Learning from Japan: Strengthening US emergency care and disaster response. *Health Affairs*, 32(12), 2172–2178. <https://doi.org/10.1377/hlthaff.2013.0704>
- Nie, H., Tang, S. Y., Lau, W. B., Zhang, J. C., Jiang, Y. W., Lopez, B. L., Ma, X. L., Cao, Y., & Christopher, T. A. (2011). Triage during the week of the Sichuan earthquake: A review of utilized patient triage, care, and disposition procedures. *Injury*, 42(5), 515–520. <https://doi.org/10.1016/j.injury.2010.01.113>
- Li, X.-H., & Zheng, J.-C. (2014). Efficient post-disaster patient transportation and transfer: experiences and lessons learned in emergency medical rescue in Aceh after the 2004 Asian tsunami. *Military Medicine*, 179(8), 913–919. <https://doi.org/10.7205/MILMED-D-13-00525>
- Ardagh, M. W., Richardson, S. K., Robinson, V., Than, M., Gee, P., Henderson, S., Khodaverdi, L., McKie, J., Robertson, G., Schroeder, P. P., & Deely, J. M. (2012). The initial health-system response to the earthquake in Christchurch, New Zealand, in February, 2011. *The Lancet*, 379(9831), 2109–2115. [https://doi.org/10.1016/S0140-6736\(12\)60313-4](https://doi.org/10.1016/S0140-6736(12)60313-4)
- Kelen, G. D., McCarthy, M. L., Kraus, C. K., Ding, R., Hsu, E. B., Li, G., Shahan, J. B., Scheulen, J. J., & Green, G. B. (2009). Creation of Surge Capacity by Early Discharge of Hospitalized Patients at Low Risk for Untoward Events. *Disaster Medicine and Public Health Preparedness*, 3(S1), S10–S16.
- Awais, S., & Saeed, A. (2013). Study of the severity of musculoskeletal injuries and triage during the 2005 Pakistan earthquake. *International Orthopaedics*, 37(8), 1443–1447. <https://doi.org/10.1007/s00264-013-2013-3>

Response #3c: how practical it is to assume that the key hospital resources (such as ambulance) can be shared freely across multiple cities OR deployment of additional EMTs and the availability of resources

(e.g., space, medical professionals) when main facilities are already overwhelmed and crowded. Some discussion is needed on the state-of-practice, placement protocols, etc.

Response #3c:

In terms of ambulance sharing, the Peruvian Health Ministry (MINSA) and the Social Security (Essalud), the institutions that manage the hospitals in our case study, have launched emergency units that aim to coordinate the response of ambulances. MINSA has the “Servicio de Atención Móvil de Urgencia” (SAMU) and Essalud has the “Sistema de Transporte Asistido de Emergencia” (STAE). While SAMU and STAE manage a reduced number of ambulances to severe injuries on a regular basis, they have capabilities to coordinate at the system level with multiple hospitals from MINSA and Essalud, respectively. To deploy a highly coordinated earthquake emergency response, SAMU and STAE will need to enhance their communication capacities with software and hardware to manage a joint system response, closely monitoring ambulance and operating room usage and also tracking patient arrivals, triaging, treatment and transfers. Currently, the emergency response to COVID19 is being led by SAMU from MINSA. SAMU and MINSA are enhancing communication capacities at the system level on the fly to effectively coordinate the use of mechanical ventilators and COVID19 testing equipment with hospitals from both MINSA and Essalud. The emergency response research community will have an enormous opportunity to learn from the COVID19 emergency response in Peru (and worldwide) to propose novel solutions to ensure healthcare demands are met by the hospital capacity for the next emergency responses.

In other countries, there are more capabilities for a coordinated ambulance response already in place. For example, in the U.S., past earthquakes showed evidence of system-level coordination for emergency response. In the county of Los Angeles, the Los Angeles County Emergency Medical Services Agency led and coordinated the system-level response after the 1994 Northridge Earthquake, which caused around 10,000 people injured (Shoaf et al., 1998). During the response, the Agency coordinated with 76 hospitals using communication by radio. The agency assessed the functionality of hospitals, evaluated which healthcare facilities were overloaded, and coordinated patient mobilization at the county level effectively (Stratton et al., 1996).

Therefore, it is extremely important to build capabilities for high coordination during an emergency response. Past experiences have shown partial success in such coordinated efforts, thus, cities and countries should enhance their coordination and communication capabilities with the acquisition of robust hardware (e.g., radios, satellite cell phones) and software (e.g., that can track patients and resources during an emergency) and with further drills and training for the hospital personnel. Our results help understand which hospitals are more likely to send and receive patients, thus initial interventions to enhance coordination capabilities should focus on such high-priority hospitals.

We included a summary of this point in line 362 as follows:

Emergency response agencies under MINSA and Essalud, “Servicio de Atención Móvil de Urgencia” (SAMU) and “Sistema de Transporte Asistido de Emergencia” (STAE), respectively, have capacities to coordinate across multiple hospitals on a system level. SAMU and STAE must ensure the continuity of their existing communication capacities and enhance their coordination capacities to be able to direct the hospitals' ambulances strategically in Lima during an emergency response. Acquisition of hardware such as satellite cell phones and radios for all ambulances and hospitals and implementation of robust software to manage and process the patient treatment, triaging, and transfer information will be key to accomplishing such levels of coordination. Past

earthquakes have shown that emergency responses with high coordination are feasible even with older technology. After the 1994 Northridge earthquake, the Los Angeles County Emergency Medical Services Agency coordinated with 76 hospitals using radio communications. The agency assessed the functionality of hospitals, evaluated which healthcare facilities were overloaded, and coordinated patient mobilization at the county level effectively [57].

In terms of the deployment of mobile operating rooms by EMTs, past earthquakes showed evidence of the successful deployment of such units by either national or international agencies. We added the following explanation to the paper in line 384:

Finally, our findings also show how EMTs can strategically deploy mobile operating rooms leveraging high coordination capacities and system-level information. In past earthquakes, EMTs have been key to deploying such life-saving resources worldwide [25], e.g., 1999 Turkey earthquake [62], 2004 Sumatra earthquake [63], 2010 Haiti earthquake [64]. Using similar hardware and software requirements for system-level coordination to the third strategy, the fourth strategy can inform where and how many of the 15 mobile operating rooms should be deployed by EMTs, further demonstrating that by supplying the periphery with additional medical resources, the city will be more successful at decreasing global patient waiting times and offloading demands for patient mobilization. Without coordination, EMTs might find it practical to locate these additional operating rooms at the city center, where equipment mobilization is easier, and the availability of doctors and nurses is higher. However, such a plan will lessen the ability of the hospital system to effectively treat patients because more patients would have to be transferred from the periphery to the center, overloading the roads and potentially overflowing ambulance capacities.

As previously discussed, we also added a description of the time to deploy the mobile operating rooms and their locations in close proximity to existing hospitals in line 252 (same as in response to comment 3a):

We assume that the additional operating rooms will be functioning three days after the earthquake. Though three days can be a short time to deploy additional operating rooms, well-organized surgical teams have demonstrated that they can deploy their medical resources even within 48 hours after the disaster even from abroad, e.g., 2007 Pisco earthquake in Peru [46], 1999 Turkey earthquake [47]. This strategy deploys the operating rooms close to existing hospitals to leverage their triage areas and additional resources such as personnel, power generators or backup water. Such arrangements worked well during the emergency response following the 2010 Chile Earthquake and extended the existing capacity of healthcare facilities [4].

References

- Shoaf, K. I., Sareen, H. R., Nguyen, L. H., & Bourque, L. B. (1998). Injuries as a result of California earthquakes in the past decade. *Disasters*, 22(3), 218–235. <https://doi.org/10.1111/1467-7717.00088>
- Stratton, S. J., Hastings, V. P., Isbell, D., Celentano, J., Ascarrunz, M., Gunter, C. S., & Betance, J. (1996). The 1994 Northridge earthquake disaster response: The local emergency medical services agency experience. *Prehospital and Disaster Medicine*, 11(3), 172–179. <https://doi.org/10.1017/S1049023X00042916>

Comment #4: The mean and 90th percentile estimate of city-wide waiting times between strategy 3 & 4 are remarkable close (10-19 & 8-15 days). It is not entirely clear whether in strategy 4 authors assumed the ambulance sharing option was available or not (line 249?). The authors need to tabulate all the results clearly.

Response #4:

With strategy 4, the ambulance sharing option is also available. We clarified this in line 261:

Additionally, with the fourth strategy hospitals can share ambulance capacities across the city, analogously to the emergency response with the third strategy.

The only difference between strategy 3 and 4 is that 15 operating rooms (ORs) are added to the system with the latter strategy. The waiting times do not decrease more because 15 ORs only represent 17% of the average number of functional ORs (87) in Lima after the earthquake (see line 130). Thus, the difference between the expected waiting times with the fourth and third strategies is not as high and close to the increase in the OR capacity (~20%). We clarified this point in line 276:

With this strategy, only 15 additional operating rooms were deployed, representing only a 17% increase with respect to the mean number of functioning operating rooms in the entire city after the earthquake. If the EMTs deploy more operating rooms using this methodology, then treatment times will be further reduced, the periphery will be better supplied with needed resources, and the usage of ambulances and critical roads will be further offloaded in the city.

Moreover, to make sure we addressed your comment properly, we revised all our datasets and scripts. We determined that all the datasets and scripts were running as expected except for the location of Hospital de Ventanilla. Previously, we located the hospital in the correct region but with the incorrect geographical coordinates. We updated this information and reran all the assessments. The results barely changed (< 10% in global waiting times for all strategies) as the hospital had been located in the correct region before. We updated the data in the repository (see the Data Availability section).

Comment #5: Discussion section need a lot of work. The authors need to emphasize on significance of each strategy, their practicability, significance of results, alternate ways for improvements in enhancing capacity, limitations of proposed method, practical constraints, and scope of future research more clearly.

Response #5:

Thank you for pointing this out. By addressing your previous comments, we have iterated the discussion section significantly. The following explains how we specifically addressed the main two concerns.

1) Significance of each strategy, results and their practicality: As pointed out in the responses to your comment 3, we have included descriptions addressing the significance and practicality of each strategy based on emergency responses to previous earthquakes. Specifically, in the new version of the Discussion section, we explain:

- a) Triage procedures in previous earthquakes to identify high-severity injuries
- b) The “reverse triage” procedure to offload demands in hospitals
- c) Feasibility and potential implementation of low-coordination strategies

- d) Feasibility and potential implementation of high-coordination strategies
- e) The potential declaration of “State of Emergencies” in case of this earthquake and its relationship to the deployment of goods and services
- f) SAMU and STAE as potential agencies to lead the emergency response with high coordination (with example in the 1994 Northridge earthquake)
- g) Deployment of field operating rooms and EMTs in previous earthquakes

2) Limitations of the proposed method and practical constraints, and scope of future work: We added a complete subsection in line 610 within the Methods section describing the limitations of our work and how future research can use our formulation as groundwork to address it. We added the following:

3.5. Model limitations and future work

Our formulation and case study advance the field of earthquake emergency response. However, there are existing limitations that must be addressed in future research. First, we model patient transfers that heavily rely on the proper functioning of the transportation system during the emergency response. However, our paper does not capture that the transportation network can be disrupted due to potential bridge failures, seismic liquefaction, or infrastructure debris caused by the earthquake. While there are models to capture these disruptions in a large city, we did not capture them because they require extensive seismic vulnerability data currently not available for Lima. Only a few large urban centers worldwide have sufficient vulnerability data to start capturing disruptions in the entire transportation network, e.g., the Bay Area in California [69,88]. Because collecting such data often requires significant resources from cities, strategic surveying of critical elements (e.g., critical bridges and roads) in the system can be an effective starting point for cities that lack sufficient vulnerability data. Our results can be used to support such a strategic surveying approach in large urban centers. Instead of exhaustively collecting the vulnerability data of all components of the transportation network, the de facto approach in risk analysis, researchers can use our results to prioritize surveying the critical roads for patient transfers to evaluate its potential disruptions due to structural failure, soil liquefaction, and debris blockage and propose strategic risk mitigation measures for the network. Similarly, the power, water, and sewage networks [89,90] are pivotal for the proper functioning of the hospital facilities. Our results can help future research focus on the power substations, power distribution lines, and water and sewage pipes serving the critical hospitals in the system. Focusing on these critical components will enable large urban centers that lack data to have a robust starting point for including the interactions between the hospital system and other critical urban systems.

Second, our case study only included an evaluation of an earthquake occurring at nighttime because we lacked data with the spatial distribution of the urban population at other times of the day. The spatial distribution is an important factor in the assessment of the spatial distributions of earthquake casualties because it enables us to track what vulnerable buildings are occupied. In our nighttime scenario, people are mostly at residential buildings, which are particularly vulnerable in Lima. We used LandScan information, which is available worldwide [91], as a proxy for the nighttime distribution of the population [19,20]. Earthquake scenarios occurring during the daytime, will have a different distribution of population and therefore a different spatial distribution of earthquake casualties. While we were not able to track the time variations of the urban population in Lima during day and night, existing work has already shown that mobile network data can be effectively used for it, e.g., Beijing in China [92]; Turin, Rome, and Milan in

Italy [93]. As more mobility data are rapidly collected by public and private institutions, plenty of opportunities to create large databases with time-variant high-resolution urban densities will be available in the near future. Deployments of models to track these time-variant urban densities will enable researchers to leverage the formulation presented here and extend the emergency response application to different times of the day in multiple cities so that emergency managers are better informed on how to protect their communities.

We also added a briefer description of the limitations and potential future work in the Discussion section in line 399:

Though this paper provides novel understanding on the spatial mismatch between capacity and demand and describes strategies to address it, more research is still needed to understand how the vulnerability of other critical urban systems, such as the transportation or power system, play a role during the emergency response of the city. Robust methods for capturing the role of these urban systems have already been developed [65,66]; however, lack of exhaustive datasets with exposure and vulnerability information make such analyses prohibitive in most cities. Our emergency response assessment can help direct data collection efforts to hotspots in the city, such as to specific roads that are critical for emergency response, or specific areas where the city must ensure that the power supply is sustained or rapidly recovered. Effective plans will capitalize on the methodology and information provided here to better prepare cities facing high significant risk from future large earthquakes.

Reviewer #2 (Remarks to the Author):

The paper "Effective Plans for Hospital System Response to Earthquake Emergencies" is an interesting work, but I don't think that it's appropriate for Nature Communications publication. I don't see what the new discovery or significant result of general interest is that would make this appropriate for Nature Communications. I'm sure there are more specialized journals that this would be appropriate for.

Dear reviewer #2,

We are grateful that you took the time to review our work. We appreciate the insightfulness of your comments, suggestions, and constructive criticism. We modified the paper according to your comments and suggestions, and as a result, the paper is in much better shape than its initial version. By addressing your comments and suggestions, we have described more clearly the novelty and contributions of our work. We thank you for helping us strengthen our paper. Hereafter, we provide detailed responses to your specific comments.

Comment #1a: The authors say that "a methodology to evaluate emergency response" is presented and that this proposed methodology "is able to design effective plans for patient transferal and allocation of medical resources". In the paper though this methodology is only used to evaluate four **very simplistic emergency response** plans the outcomes of which are obvious.

Response #1a:

Thank you for this comment. We acknowledge that the four emergency response strategies are conceptually simple, e.g., two very low-coordination strategies and two very high-coordination strategies. We deliberately choose these conceptually simple strategies to show very clearly how high and low-coordination capacities perform during an emergency response and utilize the results to highlight the key system features that make the latter strategies work better than the former ones in very large urban centers.

Nevertheless, the conceptual simplicity of the strategies should not be extrapolated to the challenge of solving specific protocols for each hospital in a large urban center. There is a high level of complexity in finding the specific transfers, triage, and treatment protocols for each hospital. As discussed in the paper, Lima is a megacity that should expect a large number of patients with a non-uniform spatial distribution throughout the city after a magnitude 8.0 earthquake. Additionally, the hospital system is quite complex with 41 healthcare facilities distributed through the city that have different quantities of medical resources and different levels of seismic vulnerabilities.

Under these conditions, the decision-making process for individual hospitals becomes quite complex even with the conceptually simple global strategies presented in this paper. Due to the complexity and large extent of Lima, hospitals' decisions of transfer, treatment and triage will have system-level implications, i.e., too many transfers to one specific hospital can lead to a capacity overflow and affect other parts of the system (e.g., ambulance usage or even roads). Thus, to properly inform earthquake response in large urban centers, a formulation that can handle such complexities and can represent hospitals as a system rather than as individual units is needed.

Our formulation handles such complexities at the system-level and can inform hospitals on effective strategies to mobilize and treat patients to minimize global waiting times. Our methodology captures the non-uniform spatial distributions of healthcare demand and capacity, and we show that it is able to find the best transfer, triage, and treatment strategies for each of the 41 hospitals in the city and for a time horizon of 100 days after the earthquake using 1-day time steps. To the best of the authors' knowledge, there is no existing formulation that can solve for such specific emergency response protocols after an earthquake with this level of detail.

We are grateful that you raised this point. We understand that we did not convey the complexity of the problem in the best way before, so we added the following sentences to our Discussion section in line 321:

We chose four strategies due to their conceptual simplicity to represent two scenarios with low-coordination capacities and other two with high-coordination capacities. While simple conceptually, these strategies involve quite complex specific protocols for each hospital that need to be adjusted according to the hospitals' resources and the number of remaining patients in triage areas. Nevertheless, we show that our proposed formulation can find effective protocols under the high levels of complexity of the earthquake emergency response in Lima.

Comment #1b: In order to apply this methodology the "unique" dataset of high-resolution hospital vulnerability is needed and that might not be generally available.

Response #1b:

This point has enormous relevance to the application of the methodology in other cities. While the hospital vulnerability data might not be generally available worldwide, datasets with good potential to be used with our formulation do exist in several cities as part of previous or ongoing earthquake risk studies.

The Disaster Risk Management group (DRM - <https://www.worldbank.org/en/topic/disasterriskmanagement>) and the Global Facility for Disaster Risk Reduction (GFDRR - <https://www.gfdr.org/en>) from the World Bank are very active and experienced in collecting such large datasets for critical infrastructures, e.g., schools in Colombia, El Salvador, Uzbekistan, Kyrgyz Republic (for example see the "Glosi" database in Global Program for Safer Schools - GPSS, <https://gpss.worldbank.org/en/glosi>). The lead author has worked with the World Bank on some of these projects. In fact, in a recent project with the World Bank, the lead author worked on processing exposure data and performing risk analysis for hospitals in Uzbekistan. While some preprocessing effort will be needed to merge the hospital vulnerability information with the hospital capacity information as we did in our paper in Lima (e.g., operating rooms, ambulances), we consider that the data collected in the World Bank projects will provide a robust starting point to apply our assessment to other countries, e.g., in Uzbekistan (<https://www.worldbank.org/en/news/feature/2016/12/14/uzbekistan-moves-towards-proactive-approach-to-disaster-risk-management>).

Moreover, our formulation incentivizes the collection of exposure data in other seismic countries. Traditionally, city-wide vulnerability data has been used with a heavy focus on computing risk metrics associated with financial loss, for example, the average annual loss to repair or replace damaged infrastructure after earthquakes. This is the case of the dataset in Lima, Peru [Santa Cruz et al. 2012]. It was initially used to evaluate only the average annual loss of the portfolio in 2012. Our paper shows that if countries and cities collect such datasets, they can use the data to inform their emergency managers on

strategic protocols to treat earthquake patients in an effective and timely manner, especially in very large and complex urban centers. Thus, we see our paper as a piece of evidence that will inform the DRM and GFDRR groups on how these data with hospital vulnerability can be used to inform for emergency response in addition to the traditional evaluation of financial losses.

Finally, the authors are currently working on applying our methodology to the Bay Area for earthquake risk and also extending the assessment to hurricane risk in New York City. Thus, we consider that our methodology will be applied to other cities shortly.

References

- Santa Cruz, S., Blondet, M., Muñoz, A., Palomino Bendezú, J., & Tamayo, R. (2013). Evaluación Probabilística de riesgo Sísmico de Escuelas y Hospitales de la ciudad de Lima. Componente 2: Evaluación Probabilística del Riesgo Sísmico de Hospitales en la ciudad de Lima.

Comment #1c: Also, the proposed metric is based on patient waiting times for the emergency room use and effective use of ambulance patient transfers, but there are other important parameters, like the number of patients a hospital can hold in general.

Response #1c:

We appreciate that the reviewer highlights the missing parameters in our model, specifically referring to “number of patients a hospital can hold in general”. To answer the reviewer’s specific question, we clarify that our understanding of the reviewer’s specific point of “number of patients a hospital can hold in general” refers to the capacity of the hospital to treat other patients with less severe injuries who will not require surgical procedures, so we formulated an answer with this interpretation of the reviewer’s comment.

In our paper, we only focused on critically injured patients who will need surgical procedures during the emergency response. Therefore, our resilience metric is based on the time that these critically injured patients will have to wait until receiving treatment in operating rooms. We deliberately narrowed down the scope of the paper to these high-severity injuries due to two main reasons.

First, high-severity patients will be a priority for treatment in the healthcare system during the emergency response so we considered that the hospitals will use most of their resources to treat these critically injured patients first. Past earthquakes have shown that emergency responders triaged patients into a few classifications and gave very high priority to these high-severity injuries. For example, after the 2008 Wenchuan earthquake in China, the 2283 earthquake patients at the West China Hospital of Sichuan University, the largest hospitals of Sichuan province, were triaged into four main groups: resuscitation, urgent treatment, delayed treatment, and minor injuries (Nie et al., 2011), which allowed the doctors to identify patients needing immediate treatment. Similar triaging procedures with a few categories also occurred in other countries, e.g., in Indonesia after the 2004 Sumatra earthquake (Li and Zheng, 2014), in New Zealand after the 2011 Christchurch earthquake (Ardagh et al., 2012). Additionally, hospitals apply the principle of **reverse triage** (Kelen, 2009). Under a reverse triage protocol, hospitals offload the facility of the most minor wounded patients and give more attention to the seriously injured patients, e.g., 2005 Pakistan earthquake (Awais, 2013), or even after hurricanes in Sandy [Wang2017].

Additionally, we identified that a key bottleneck in the treatment process of these critically injured patients is the availability and proper functionality of operating rooms. Many critically injured patients will need beds to recover after undergoing surgical procedures. However, we did not include the number of beds in our assessment because the existing hospitals' capacity of beds is significantly larger than for operating rooms (10,070 hospital beds vs 182 operating rooms in the system). Therefore, by focusing on patients who will need surgical procedures and having as metric waiting times for the operating rooms, our model should be properly representing that the emergency responders will give priority to high-severity patients and that the bottleneck to treat them is in the functionality of operating rooms.

We added a paragraph in line 300 to discuss this point:

We found that a M 8.0 earthquake in Lima will cause an emergency response with 4.7k critically injured people requiring surgical procedures, which will impose large demands on a hospital system that will work only at 48% capacity due to earthquake disruptions. As in previous emergency responses, triage procedures will be key to identifying these critically injured patients and to enable doctors to give them priority for treatment, e.g., 2004 Sumatra earthquake in Indonesia [48], 2008 Wenchuan earthquake in China [49], 2011 Christchurch earthquake in New Zealand [50]. Additionally, the activation of "reverse triage" protocols in Lima can help offload demands by releasing minor wounded patients to focus on these critically injured patients as it happened in the 2005 Pakistan earthquake [51] or even in Hurricane Sandy [52].

Second, we choose to focus on high-severity injuries to be able to put together the fundamental pieces of the model emergency response. We connected a demand and capacity assessment of healthcare with a dynamic model of the hospital system for emergency response. By capturing the capacity of each hospital, we are explicitly evaluating the number of patients that each hospital can treat in operating rooms (and also mobilize). This model is unique in the sense that, to the best of our knowledge, existing literature has not looked at the system-level behavior of emergency response, especially for very large urban centers. Therefore, we see our model as the groundwork for future studies that will consider multiple patient severities and multiple treatment procedures, i.e., "the number of patients a hospital can hold in general". One direction for future work is to extend the hospital system model with a multi-commodity network flow model (Fenorova, 2010) to capture such different injury severities and respective treatment processes.

References:

- Fonoberova, M. (2010). Algorithms for Finding Optimal Flows in Dynamic Networks. In S. Rebennack, P. M. Pardalos, M. V. F. Pereira, & N. A. Iliadis (Eds.), Handbook of Power Systems II. Springer. <https://doi.org/10.1007/978-3-642-12686-4>

Comment #1d: One more shortcoming of this study is that the authors consider only a nighttime scenario, which significantly reduces the generality of the approach. During the daytime, many people who live in the peripheral regions might be in the center of the city.

Response #1d:

This point is quite relevant and worthy of careful discussion. We advocate strongly for the generalization of the applicability of our framework but advocate only partially for the generalization of our specific results. First, we discuss the generalization of our framework, and then we discuss the generalization of the specific results.

In terms of the generalization of our formulation, we have studied the problem of earthquake emergency response thoroughly and propose a formulation that incorporates its three key factors: the surge in demand through a probabilistic model that relates earthquake shaking with multi-severity casualties, reduction in capacity through damage to the hospital infrastructure, and system behavior that captures the ability of the healthcare system to share resources, coordinate, and deploy mobile field hospitals in different areas of a large city. The resolution of the proposed formulation is consistent with data that can be collected as part of World Bank projects (as mentioned in the response to the previous comment), municipalities, and healthcare institutions for very large cities. Therefore, our analytical framework and workflow are intended to be generalizable and deployable in different seismic regions in the world.

Please, see the sentences starting in line 61 that clarifies this point:

To demonstrate the benefits of organized, system-level responses, we present findings from applying a methodology that characterizes the disaster emergency response of hospitals as a system. Our integrative methodology combines models of multiseverity earthquake casualty estimation [19,20] and post-earthquake hospital functionality with a proposed network flow model. The network flow model captures the system behavior of emergency responses through the evaluation of patient treatment, triage process, and patient transfers across a large number of hospitals, which establishes a fundamental difference from existing formulations focused on analyses on a single-hospital scale.

Also, please, see the sentences starting in line 287:

Because our methodology considers hospitals in a large urban center to behave as a system, emergency managers and resilience officers can apply our methodology to a whole large city and evaluate optimal patient transfer strategies between hospitals, effectively allocate ambulances in the system, and guide the deployment of field hospitals.

Now, turning to the generalization of the specific results from nighttime to other times of the day, we did not choose the nighttime scenario because the results could be generalized to the daytime scenario. Instead, we chose it because it is a robust starting point for creating emergency response strategies. Because earthquakes have a uniform distribution of occurrence during the time of the day, the nighttime scenario has a slightly larger probability of occurring than the two other important scenarios: commuting time scenario (e.g., from 6 am to 8 am and from 4 pm to 8 pm) and the working time scenario (e.g., from 8 am to 4 pm). Additionally, the nighttime scenario in Lima is a potential worst-case scenario. At nighttime, people are mostly at their residential buildings, and these buildings in Lima can be very vulnerable. At the periphery of the city, these buildings are mostly confined masonry houses that are becoming even more vulnerable as families keep adding stories to the buildings without properly retrofitting them, as it happens in multiple other countries with high seismic hazards (Lallemant et al. 2017). We added a paragraph in line 125 to discuss this point:

Our results are designed for an earthquake occurring at nighttime. This scenario represents a potential worst-case scenario because it is when most people are inside residential buildings,

which are particularly vulnerable in Lima. Predominantly the city's periphery has vulnerable residential infrastructure as a result of poor construction practices and lack of seismic code enforcement [40,41]. Figure 3 shows the spatial distribution of the average number of patients who will require surgical procedures for the nighttime scenario. A comparison with the spatial distribution of nighttime population density in Lima (Figure 9 in the Supplemental Information) indicates that many of these patients are located in high-density zones. Because earthquakes have a uniform distribution of occurrence during a day, this scenario has a slightly higher likelihood of occurring than the other two important earthquake scenarios to consider: commuting hours from 6 am to 8 am and from 4 pm to 8 pm, and working hours from 8 am to 4 pm [19,20]. The Methods section includes a description of the data requirements to fully evaluate scenarios at other times with our formulation, and the Supplemental Information includes a discussion of their potential results (see Figure 11).

Thus, we do not intend to fully generalize our nighttime results to all times of the day. However, we agree with the reviewer that it is important to discuss emergency response during other times of the day, so we discuss how a potential response during daytime might look like and what are the data requirements to evaluate these scenarios with our proposed framework.

The spatial distribution of earthquake patients is not solely dependent on where the population is located at the time of the earthquake. The spatial distribution is a combination of multiple factors, including the vulnerability of the infrastructure and its spatial distribution, the earthquake shaking (including soil conditions that can amplify the shaking quite significantly), and the aforementioned spatial distribution of people at the time of the earthquake. Our previous studies in Ceferino, Kiremidjian, and Deierlein (2018a and 2018b) explain in thorough detail how these factors play a role in the surge in casualties after an earthquake. While earthquake shaking and the vulnerability of infrastructure do not change by the day, the distribution of people does, as the reviewer insightfully points out. Our previous studies highlight that it is very important to track such changes at least during three very distinctive scenarios: nighttime, working hours, and commuting times. However, our previous studies also highlight that currently, such information is not available in Lima at the resolution and to the extent needed to estimate multi-severity earthquake casualties. To estimate such a time-variant distribution of people, we need data on population mobilization (e.g., from mobile network data) that can support the creation of models to track urban density changes with high resolution to determine building and infrastructure occupancy levels for multiple times at an entire regional scale. Yet, currently, only a few cities have been able to use such information and capture these dynamic urban effects (e.g., Beijing in China [Wang et al., 2018]; Turin, Rome, and Milan in Italy [Khodabandelou et al., 2018]). This information is particularly missing in rapidly growing cities in developing countries, such as Lima. However, the collection of mobility data is rapidly growing worldwide, so plenty of opportunities to create large databases with time-variant high-resolution urban densities will be available in the near future. Deployments of models to track these time-variant urban densities will enable representations of emergency response at different times of the day. We included a discussion regarding this point in line 611:

Our formulation and case study advance the field of earthquake emergency response. However, there are existing limitations that must be addressed in future research. First, [...] Second, our case study only included an evaluation of an earthquake occurring at nighttime because we lacked data with the spatial distribution of the urban population at other times of the day. The spatial distribution is an important factor in the assessment of the spatial distributions of earthquake casualties because it enables us to track what vulnerable buildings are occupied. In our nighttime scenario, people are mostly at residential buildings, which are particularly

vulnerable in Lima. We used LandScan information, which is available worldwide [91], as a proxy for the nighttime distribution of the population [19,20]. Earthquake scenarios occurring during the daytime, will have a different distribution of population and therefore a different spatial distribution of earthquake casualties. While we were not able to track the time variations of the urban population in Lima during day and night, existing work has already shown that mobile network data can be effectively used for it, e.g., Beijing in China [92]; Turin, Rome, and Milan in Italy [93]. As more mobility data are rapidly collected by public and private institutions, plenty of opportunities to create large databases with time-variant high-resolution urban densities will be available in the near future. Deployments of models to track these time-variant urban densities will enable researchers to leverage the formulation presented here and extend the emergency response application to different times of the day in multiple cities so that emergency managers are better informed on how to protect their communities.

Finally, as discussed previously, the mobilization of people is only one of the factors in the assessment of casualty surge and emergency response. Besides, we need to capture the spatial distribution of ground shaking and the infrastructure vulnerability, which can be considered constant during a day. To detail further the coupled effect of both the distribution of ground shaking and the vulnerability of the infrastructure, we discuss the results in Figure 11 in the Supplemental Information (previously Figure 3b), and it is also shown here:

Figure 11. Spatial distribution of injury rates estimated as the mean number of earthquake patients who will need surgical procedures and the total number of people per km² for the M 8.0 earthquake. The centric districts, Lima, Breña, La Victoria, and Jesús María, where the healthcare resources are heavily concentrated. The intervals in the plot represent quantiles (20th-percentile increments) on the spatial data.

The plot is showing the injury rate per square kilometers resulting from dividing the mean number of patients who will require surgical procedures and the total population there. We use these estimates to discuss why we would expect a mismatch between demand and capacity even during other times of the day. Please, see line 676 for a discussion on it:

The distributions of casualties shown in Figures 10a and 10b result from the combination of multiple factors, including the vulnerability of the infrastructure and its spatial distribution, the earthquake shaking including soil conditions that can amplify it quite significantly, and the spatial distribution of people at the time of the earthquake (Figure 9). While earthquake shaking and the vulnerability of infrastructure do not change at different times of the day, the spatial distribution

of people does, thus the distribution of casualties will be different if the earthquake occurs during the daytime. Our previous studies highlight that it is very important to track such changes at least during other two very distinctive scenarios: working hours and commuting times [19,20].

As explained in the Methods section, lack of data did not enable us to explicitly model these two scenarios; however, we present Figure 11 to discuss their potential casualty distribution scenarios. The plot shows a normalized spatial distribution of patients who will need surgical procedures. It is estimated as the division between the mean number of patients who will require surgical procedures and the total population at nighttime per km². Figure 11 shows that the patient rates in the four centric districts are close to the median values of the entire city. The average estimate across these four districts is 0.48 patients per thousand people, i.e., there are 490 patients and ~1 M people there at the time of the earthquake. Outside these four districts, the rates have large variability. In multiple areas in the periphery, there are high rates, with some hotspots reaching values up to 1 or 2 patients per thousand people. These high rates are a result of significant building vulnerabilities and poor soil conditions in the region, especially in the coastal districts, e.g., softer sands in Villa El Salvador and softer clays in La Punta. Yet, the average rate outside the centric districts is 0.52 patients per thousand people, only 7% larger than in the centric districts.

Assessments aiming to evaluate the surge in casualties during the day can combine the population distribution and building occupancy during such hours with these rates to understand the distributions of healthcare demand and capacity at different times. We expect people to mobilize to the center of the city during the day. While this mobilization might help reduce the number of casualties in the periphery, we do not expect this reduction to solve the mismatch between demands and capacity. If we perform a simple linear extrapolation with the estimated rates, we observe that 40% of the total population in Lima would need to travel to the four centric districts during the day so that there is no mismatch between demands and capacity. While lack of data did not enable us to track the increase in density during the daytime in these four districts, we were able to use census information to estimate an upper bound on the number of people in these districts during the daytime. The census information in Lima shows that only 25% of the population in Lima works in different districts from where they live. While not all these people will go to work to these four centric districts, even if they do, Lima would still face a mismatch between demands and capacities during other times of the day as a result of the heavy concentration of healthcare resources in the center of the city.

In summary, while so far we have not considered the daytime scenario in full detail, the nighttime scenario is a robust starting point for creating emergency response strategies given that it can be a potential worst-case scenario and that it has a slightly higher likelihood of occurring than the two other relevant scenarios, i.e., commuting or working times. We provide results to create policies for the nighttime scenario and a comprehensive approach to evaluate other times of the day once mobilization data are available. While we cannot fully generalize the results to other times of the day, we could generalize them for other cities for nighttime scenarios. Other cities in Latin America are also characterized by poor construction practices, especially from low-income families who live at the periphery of the city. If these urban patterns exist in a city, then it is very likely that mismatches between the capacity and demand of healthcare services after an earthquake will arise in an earthquake emergency response.

References:

- Lallemand, D., Burton, H., Ceferino, L., Bullock, Z., & Kiremidjian, A. (2017). A Framework and Case Study for Earthquake Vulnerability Assessment of Incrementally Expanding Buildings. *Earthquake Spectra*, 33(4), 1369–1384. <https://doi.org/10.1193/011116EQS010M>
- Wang, D., Zhong, W., Yin, Z., Xie, D., & Luo, X. (2018). Spatio-temporal Dynamics of Population in Shanghai: A Case Study Based on Cell Phone Signaling Data. In *Big Data Support of Urban Planning and Management* (pp. 239–254). Springer, Cham.
- Khodabandelou, G., Gauthier, V., Fiore, M., & A. El Yacoubi, M. (2018). Estimation of Static and Dynamic Urban Populations with Mobile Network Metadata. *IEEE Transactions on Mobile Computing*. <https://doi.org/10.1109/TMC.2018.2871156>

Comment #2:

It appears that a motivation of the paper is that cities are encouraged by relevant authorities to reach some standard for being prepared for providing medical treatment after an earthquake, and cities can follow the approach described in the paper to compare the quality of the medical care they could provide if they follow each of the scenarios described.

As pointed out by the reviewer and as discussed in the paper, the motivation of the paper is to help strategize effective earthquake emergency responses in large cities. Our results show that a large number of injured people will require surgical procedures. They also show the extent to which operating rooms might be lost after an earthquake and how local demands and capacities will be unbalanced, leaving the periphery in Lima particularly vulnerable. The paper also aims to inform emergency responders how much reductions in waiting times they should expect with high-coordination responses. Additionally, the paper provides a tool that can optimize and solve for specific triage, treatment, and transportation strategies for each hospital. The tool can also be used to strategically distribute additional resources (e.g., mobile operating rooms) in specific locations and quantities in the system to minimize waiting times. As discussed previously, the tool has been applied to Lima, Peru, for a nighttime scenario, but with the right input data, it can be deployed to other regions of the world and other times of the day. Please see line 61 (same as response to comment 1d):

To demonstrate the benefits of organized, system-level responses, we present findings from applying a methodology that characterizes the disaster emergency response of hospitals as a system. Our integrative methodology combines models of multiseverity earthquake casualty estimation [19,20] and post-earthquake hospital functionality with a proposed network flow model. The network flow model captures the system behavior of emergency responses through the evaluation of patient treatment, triage process, and patient transfers across a large number of hospitals, which establishes a fundamental difference from existing formulations focused on analyses on a single-hospital scale.

Also, in line 294 (same as response to comment 1d):

Because our methodology considers hospitals in a large urban center to behave as a system, emergency managers and resilience officers can apply our methodology to a whole large city and evaluate optimal patient transfer strategies between hospitals, effectively allocate ambulances in the system, and guide the deployment of field hospitals.

Also, the model can be used for preparation for future earthquakes or even online after an earthquake happens as soon as they emergency responders collect the information with the distribution of injuries and functional hospital resources. Please see sentences clarifying this in point in line 243:

To deploy these high-coordination strategies during the emergency response, SAMU and STAE will need a few days after the next large earthquake to collect necessary system-level information including both casualties and the residual hospital functionality [58–61]. They could use the trends shown in Figure 7c for earthquake preparedness to properly plan the pragmatic implementation of effective strategies for the next earthquake emergency responses in Lima.

Comment #2a:

Presumably one expects the more complex scenarios to be better, but perhaps the question is whether the added cost is worth the improvement in medical care or patient outcome. The paper should really have much more discussion of how to apply the approach to arbitrary cities, and maybe calculate the cost of the different scenarios to aid decision making.

Response #2a:

The reviewer points out a matter of constant debate in the disaster risk management field. The cost of risk mitigation measures and their respective benefits have been explored from multiple, but not enough, perspectives. The cost-benefit assessment of disaster risk management is a complex problem because benefits can be hard to quantify and, more importantly, the benefits cannot always be monetized. This is the case of emergency response strategies. In case of emergency response, governments will not choose a cheaper option just to save money. Instead, during an emergency response, they will use their resources to their maximum capacity and in the best way they can to save the largest amount of lives, as we are currently seeing in the response to COVID19 in Peru and multiple countries.

To access emergency funds and speed up procedures to purchase goods and services, governments have mechanisms such as the declaration of “State of Emergency”. Thus it is unlikely that during an emergency, the government will choose to spend less money or resources than they can at the expense of losing lives. For example, after confirming the severity of the 2007 Pisco earthquake, the Peruvian Government declared a State of Emergency in the affected region to access the respective emergency funds [Bambaren, 2010]. We added the following paragraph in line 294 to point out to the potential “State of Emergency” in the region to use the largest amount of resources Peru had:

Due to the severity of this emergency, the government will very likely trigger a declaration of “State of Emergency” as it happened after the M 8.0 2007 Pisco earthquake to facilitate the entry of international humanitarian assistance and the rapid purchase and deployment of goods and services for the emergency response [42].

However, we acknowledge that the paper lacked a discussion of what resources and technologies should be deployed for each of the strategies presented in the paper for Lima. The first three strategies use the same number of operating rooms and ambulances. Thus, there are no more expenses in terms of acquiring more operating rooms and ambulances. However, these first three strategies vary in the level of coordination that the system must have. With the first two strategies, hospitals respond mostly locally and only require pairwise coordination between hospitals if they lose all their operating rooms during the earthquake. The coordination levels for these two strategies can be achieved by having enough capabilities for radio or satellite cell phone communication in triage areas and ambulances. Most

ambulances have equipment for radio communication in Lima, so this would not add excessive extra costs. The first strategy requires an identification of the closest hospital with functioning operating rooms, so hospital directors should rapidly assess their hospitals and contact neighboring hospitals to evaluate the suitability of patient transfers. The second strategy requires hospitals to identify the healthcare institution with the largest number of patients of operating room resources, therefore, hospital directors will have to establish communications with the largest hospitals in Lima with functional operating rooms after the earthquake. According to our assessment, these hospitals will very likely be Hospital Edgardo Rebagliati or Hospital Guillermo Almenara, at the center of the city. We added a discussion of this point in line 329:

The first two strategies require low coordination capacities, and as a result, they will perform poorly because hospitals will respond mostly locally without leveraging strategic transfers. With the first strategy, hospitals will not establish transfer protocols unless they lose all their operating rooms. If they lose them, hospital directors will need to quickly communicate with the closest hospital that can conduct surgical procedures and establish reliable coordination for patient transfers using equipment for radio or satellite cell phone communications.

And also in line 327:

With the second strategy, hospitals also respond mostly locally. If a hospital loses all its operating rooms, the director will have to quickly communicate with the hospitals with the largest number of functional operating rooms in the city to establish a patient transfer procedure using a similar communication protocol to the first strategy.

While the third strategy uses the same number of ambulances and operating rooms as the first two strategies, it will require higher coordination capacities because the entire system needs to communicate. There are multiple ways to implement such high levels of coordination, but one way to implement it is to work with the “Servicio de Atención Móvil de Urgencia” (SAMU) from MINSA and the “Sistema de Transporte Asistido de Emergencia” (STAE) from Essalud. MINSA and Essalud manage the portfolio of hospitals in this case study, and they launched SAMU and STAE for patient mobilization. SAMU and STAE are required by law to coordinate emergency responses. Currently, the emergency response to COVID19 is being led by SAMU from MINSA. SAMU and MINSA are enhancing communication capacities at the system level on the fly to effectively coordinate the use of mechanical ventilators and COVID19 testing equipment with hospitals from both MINSA and Essalud. The emergency response research community will have an enormous opportunity to learn from the COVID19 emergency response in Peru (and worldwide) to propose novel solutions to ensure healthcare demands are met by the hospital capacity for the next emergency responses.

To achieve a coordinated earthquake emergency response, SAMU and STAE would have to enhance their coordination capabilities to monitor and direct treatment, triage, and transfers in the system, in a similar way that the Los Angeles County Emergency Medical Services Agency did during the 1994 Northridge earthquake but at a larger scale. SAMU and STAE have already coordination capabilities with the set of hospitals in the case study. These capabilities are currently being improved with the response to COVID19, but will likely need extra improvements with the acquisition of more hardware and software. All ambulances and hospitals will need reliable communication with the acquisition of more radio units or satellite cell phones. Additionally, SAMU and STAE will require software that manages all the data with patient arrivals, treatments, and transfers in the entire city, and that can process this information with tools such as the one proposed in this paper. With such software and hardware, SAMU and STAE would be able to give directions to each specific hospital. Thus, the added cost of deploying the third strategy

with respect to the second and first strategies is related to acquiring software and hardware to enhance coordination capacities. We added a paragraph to address this point in line 362:

Emergency response agencies under MINSa and Essalud, “Servicio de Atención Móvil de Urgencia” (SAMU) and “Sistema de Transporte Asistido de Emergencia” (STAE), respectively, have capacities to coordinate across multiple hospitals on a system level. SAMU and STAE must ensure the continuity of their existing communication capacities and enhance their coordination capacities to be able to direct the hospitals' ambulances strategically in Lima during an emergency response. Acquisition of hardware such as satellite cell phones and radios for all ambulances and hospitals and implementation of robust software to manage and process the patient treatment, triaging, and transfer information will be key to accomplishing such levels of coordination. Past earthquakes have shown that emergency responses with high coordination are feasible even with older technology. After the 1994 Northridge earthquake, the Los Angeles County Emergency Medical Services Agency coordinated with 76 hospitals using radio communications. The agency assessed the functionality of hospitals, evaluated which healthcare facilities were overloaded, and coordinated patient mobilization at the county level effectively [57].

The fourth strategy requires a similar level of coordination than the third strategy so the cost of implementing the logistics for system-level coordination would be quite similar. The fourth strategy enables the system to deploy additional field operating rooms. While the government can also deploy their mobile field hospitals, it is expected that such an emergency response scenario will trigger international assistance as it happened in other earthquake emergencies. These additional resources would have to be logged into the software managed by SAMU and STAE to be deployed effectively in the system.

Finally, our findings also show how EMTs can strategically deploy mobile operating rooms leveraging high coordination capacities and system-level information. In past earthquakes, EMTs have been key to deploying such life-saving resources worldwide [25], e.g., 1999 Turkey earthquake [62], 2004 Sumatra earthquake [63], 2010 Haiti earthquake [64]. Using similar hardware and software requirements for system-level coordination to the third strategy, the fourth strategy can inform where and how many of the 15 mobile operating rooms should be deployed by EMTs, further demonstrating that by supplying the periphery with additional medical resources, the city will be more successful at decreasing global patient waiting times and offloading demands for patient mobilization. Without coordination, EMTs might find it practical to locate these additional operating rooms at the city center, where equipment mobilization is easier, and availability of doctors and nurses is higher. However, such a plan will lessen the ability of the hospital system to effectively treat patients because more patients would have to be transferred from the periphery to the center, overloading the roads and potentially overflowing ambulance capacities.

Finally, we decided to describe the resources needed to deploy the strategy and not to estimate specific costs for them mainly because the objective of the paper is not to price specific hardware and software for emergency response, but to give concrete evidence supporting the deployment of high-coordination emergency responses to minimize waiting times and potentially save lives. Estimating specific costs would require an in-depth accounting assessment of the existing hardware and software from SAMU and STAE that can be leveraged by the strategies as well as a request of quotes from telecommunication and software firms, which are tasks that fall outside of the paper scope and in our option more are more suited for a consulting project than for this article.

Moreover, while the cost of disaster risk management can sometimes be large and prohibitive for countries, this is more likely for retrofit interventions of infrastructure than for the acquisition of software and hardware for emergency response. Additionally, when resources are not sufficient, there are still mechanisms to deploy such risk management measures. Countries often implement measures using a prioritization framework that enables them to put initial resources where they are needed the most, as often done in projects led by the World Bank. For emergency response, our paper shows that SAMU and STAE will have to transfer multiple patients from the periphery to the Hospital Almenara and Hospital Rebagliati, at the center of the city, thus they will have to start by enhancing coordinations between these two hospitals and the hospitals in the periphery. Additionally, EMTs will have to deploy field hospitals in close proximity to hospitals in the periphery. The government should start focusing on implementing all logistical requirements to deploy these resources in two or three hospitals in the periphery that are the highest priority, where most likely they will need the largest number of field operating rooms. While not all hospitals might be able to be fully incorporated in the system-level response in the short term due to lack of funding, the paper can be used also to prioritize the deployment of coordination and communication capabilities for the effective use of hospital resources, ambulances, and other critical infrastructure that supports the hospital system functionality. We added sentences in line 404 to clarify this point.

Our emergency response assessment can help direct data collection efforts to hotspots in the city, such as to specific roads that are critical for emergency response, or specific areas where the city must ensure that the power supply is sustained or rapidly recovered. Effective plans will capitalize on the methodology and information provided here to better prepare cities facing high significant risk from future large earthquakes.

Comment #3: There are a number of grammar mistakes and typos in the paper, for example:
Thank you for spotting the typos. We fixed all of them.

line 85: ETMs instead of EMTs
Fixed

line 99: extra "to" between "effectively" and "treat"
Fixed

line 439: "the an" at the end of the line
Fixed

line 458: extra "are"
Fixed

line 543: "distributions" is missing the "b"
Fixed

line 463: "treat" is repeated twice
Fixed

line 568: "the" is repeated twice
Fixed

The figures are out of order, for example Figure 9 is mentioned on line 122 before Figure 4. The subplots a and b in Figure 5 should be combined for a better visibility and easy comparison.

Thank you for pointing this out. Figure 9 is part of the Supplemental Information so it will be included in a separate file. Additionally, we tried merging Figures 5a and 5b as recommended by the reviewer, but the resulting plot looked too convoluted. We wanted to highlight in this plot that the distributions of demand and capacity do not match, so we added a note in the caption for it to be clearer.

Comment #4: The formulation itself of the network flow model is promising but I don't think that it's used to its full potential and that the whole methodology can be widely applied and thus I do not believe it is worth considering for publication at its current stage.

Response #4:

We thank you for your observation and appreciate that you consider that our model is promising. We think that the paper did not well explain the uniqueness and novelty of the case study previously and did not convey properly the complexities in the application of the network flow model.

In terms of the novelty of our work, we clarified this point in line 45:

To effectively develop measures for capacity-enhancing prioritization and resource sharing and allocation, national and regional governments require information based on robust methodologies that can characterize hospitals' emergency response as an interconnected system rather than as isolated units. However, research has not yet focused on developing such system-level methodologies and tools that can support governments to elaborate effective plans for emergency response and quantify their potential benefits in treating patients more effectively and saving lives.

Instead, recent studies have primarily focused on modeling emergency response only at single-hospital scale. [...] Lack of methods and high-resolution disaster risk data have hindered the extension of single-hospital scale analyses to system-level analyses on an urban scale. As a result, regional emergency response plans have not effectively addressed capacity-enhancing prioritization and resource sharing and allocation in hospital systems, especially in large and complex urban centers.

To demonstrate the benefits of organized, system-level responses, we present findings from applying a methodology that characterizes the disaster emergency response of hospitals as a system.

In terms of the complexities of the case study, please see the response to comment 1a:

“The conceptual simplicity of the strategies should not be extrapolated to the challenge of solving specific protocols for each hospital in a large urban center. There is a high level of complexity in finding the specific transfers, triage, and treatment protocols for each hospital, and our methodology can handle it. As discussed in the paper, Lima is a megacity that should expect a large number of patients with a non-uniform spatial distribution throughout the city after a M 8.0 earthquake. Additionally, the hospital system

is quite complex with 41 healthcare facilities distributed through the city that have different quantities of medical resources and different levels of seismic vulnerabilities. Our methodology captures these non-uniform spatial distributions of healthcare demand and capacity, and it is able to find the best transfer, triage and treatment strategies for each of the 41 hospitals in the city and for a time horizon of 100 days after the earthquakes using 1-day time steps. We are grateful that you raised this point. We understand that we did not convey the complexity of the problem in the best way before, so we modify our introduction as follows:"

These strategies are chosen due to their conceptual simplicity to represent low-coordination and high-coordination capacities. However, these strategies involve quite complex specific protocols that will require adjustments each day of the response and will be different for each hospital according to their resources and patients. Notwithstanding, we show that our proposed formulation can find effective protocols under such levels of complexity.

Reviewer #3 (Remarks to the Author):

Dear reviewer #3,

We would like to thank you for your comments and suggestions. We have highlighted the contributions of our paper as you suggested. We also addressed your other comments and responded below. We think the paper is in a more robust shape now. Thank you.

Comment #1: Please highlight the novelty of your research. The differences of your work with other researchers in this field is not clear.

We have clarified the differences of our research with previous models, particularly highlighting the ability of our model to represent the emergency response at the system level and its usefulness for informing the emergency response in large cities:

Please, see the paragraphs in line 45:

To effectively develop measures for capacity-enhancing prioritization and resource sharing and allocation, national and regional governments require information based on robust methodologies that can characterize hospitals' emergency response as an interconnected system rather than as isolated units. However, research has not yet focused on developing such system-level methodologies and tools that can support governments to elaborate effective plans for emergency response and quantify their potential benefits in treating patients more effectively and saving lives.

Instead, recent studies have primarily focused on modeling emergency response only at single-hospital scale. [...] Lack of methods and high-resolution disaster risk data have hindered the extension of single-hospital scale analyses to system-level analyses on an urban scale. As a result, regional emergency response policies have not effectively addressed capacity-enhancing prioritization and resource sharing and allocation in hospital systems, especially in large and complex urban centers.

To demonstrate the benefits of organized, system-level responses, we present findings from applying a methodology that characterizes the disaster emergency response of hospitals as a system.

Please, also see the paragraphs in line 99:

This research represents a first-cut assessment on the effectiveness of emergency response policies to inform city-scale decision-making that leads to more effective treatment of patients during an emergency response to a major earthquake.

Comment #2: Please place the "Acknowledgment" section at the end of manuscript (before reference section).

Modified according to your suggestion

Comment #3: Some new published relevant articles such as:
Capacity planning and reconfiguration for disaster-resilient health infrastructure by
Amir Hossein Aghapour et al, Journal of Building Engineering
Volume 26, November 2019.

should be included in the manuscript.

Thank you for pointing us to this paper. The paper applies optimization techniques to evaluate how health-care facilities can allocate their resources in the aftermath of an earthquake. However, unlike our proposed model, their analysis is focused on individual hospitals rather than on a system of hospitals.

We added the reference in the paper in line 54:

Other studies used emergency medicine modeling tools, such as discrete event simulation (DES) and flow models, to characterize emergency response and evaluate post-disaster resource allocation but also at a single-hospital scale [10-14].

Comment #4: I believe the revised manuscript by considering the above comments could be accepted for publication.

Thank you very much. We appreciate your comments and the time you took to review our manuscript

REVIEWERS' COMMENTS:

Reviewer #2 (Remarks to the Author):

I consider that the authors' responses adequately addressed my comments, and I can recommend the revised manuscript for publication as is.

Reviewer #3 (Remarks to the Author):

Thank you for considering my comments. The revised manuscript could be accepted for publication.